# Divergent global-scale temperature effects from identical aerosols emitted in different regions

Geeta G. Persad[1] & Ken Caldeira[1]

The distribution of anthropogenic aerosols' climate effects depends on the geographic distribution of the aerosols themselves. Yet many scientific and policy discussions ignore the role of emission location when evaluating aerosols' climate impacts. Here, we present new climate model results demonstrating divergent climate responses to a fixed amount and composition of aerosol—emulating China's present-day emissions—emitted from 8 key geopolitical regions. The aerosols' global-mean cooling effect is fourteen times greater when emitted from the highest impact emitting region (Western Europe) than from the lowest (India). Further, radiative forcing, a widely used climate response proxy, fails as an effective predictor of global-mean cooling for national-scale aerosol emissions in our simulations; global-mean forcing-to-cooling efficacy differs fivefold depending on emitting region. This suggests that climate accounting should differentiate between aerosols emitted from different countries and that aerosol emissions' evolving geographic distribution will impact the global-scale magnitude and spatial distribution of climate change.

[1] Department of Global Ecology, Carnegie Institution for Science, 260 Panama Street, Stanford, CA 94305, USA. Correspondence and requests for materials should be addressed to G.G.P. (email: gpersad@carnegiescience.edu)

The global distribution of anthropogenic aerosol emissions has evolved substantially over the industrial era. In the mid-20th century, North America and Western Europe were the primary anthropogenic aerosol source regions, whereas today South and East Asian sources dominate anthropogenic aerosol emissions[1]. Aerosol particles have direct negative impacts on air quality, human health, and agricultural productivity in the regions in which they are concentrated[2–4], but also change regional and global climate through their interactions with solar radiation and clouds. In the global total, anthropogenic aerosols are estimated to have offset approximately a third of the global-mean greenhouse gas driven warming since the 1950s[5]. However, because of aerosols' relatively short atmospheric lifetime, their atmospheric distribution and temperature effects are heterogeneous and dependent on the distribution of emissions[6]. With this geographic redistribution of aerosol emissions, therefore, comes the potential for redistribution of their climate effects—a characteristic not present with long-lived greenhouse gases.

Aerosols' heterogeneous spatial distribution is recognized to influence their overall climate impact relative to more homogeneous climate forcers, like carbon dioxide[7,8]. Although carbon dioxide radiative forcing has some spatial structure associated with the climatological radiative environment[9], the historical spatial distribution of aerosol forcing has been shown to generate a larger transient and equilibrium global-mean climate response than equivalent amounts of long-lived greenhouse gas forcing, as a result of historical aerosol forcing's greater spatial coincidence with Northern Hemisphere land and polar regions[10–12]. This difference in the rate at which aerosols' top-of-atmosphere radiative forcing produces global-mean temperature effects relative to greenhouse gases' is characterized by the efficacy of its radiative forcing[7] (i.e., global-mean temperature response divided by global-mean top-of-atmosphere radiative forcing), and is a major factor in determining the degree to which anthropogenic aerosols have offset greenhouse-gas driven warming over the historical period[10,11,13]. Localized aerosol concentrations also strongly influence regional temperature, rainfall, and circulation in emitting regions[14].

Yet, despite the acknowledged sensitivity of aerosols' total climate impacts and the spatial distribution thereof to the spatial distribution of the aerosol emissions themselves, many scientific and policy discussions treat the climate effect of aerosol emissions from all regions as equal. Policy analysis continues to use single global-mean metric values, such as Global Warming Potential, to trade off the impacts of aerosol emissions from any region against greenhouse gas emissions[15], despite studies demonstrating that, for certain aerosol species, this value may differ substantially depending on the emitting region[16,17]. The reduced-form integrated assessment models used to analyze costs and benefits of climate policy similarly treat aerosol emissions from all regions as having the same capacity to affect global climate[18]. Global-mean aerosol radiative forcing and its offsetting effects on greenhouse gas-driven warming, meanwhile, have not been widely recognized to be dependent on the evolution of global aerosol emissions' spatial distribution[11,13].

The scientific community has made great progress to date in building a theoretical framework for understanding the importance of the spatial heterogeneity in anthropogenic aerosol forcing. However, limitations remain in leveraging the existing literature to address key scientific and decision-making issues surrounding the differential role of aerosols emitted from different regions. Many studies assessing the temperature response to regionally distinct aerosol perturbations use radiative forcing or atmospheric aerosol concentrations as their unit of regional perturbation[19–21]. However, these units are not easily attributable to individual regions, as radiative forcing or atmospheric aerosol

concentrations occurring in a given region may be attributable to aerosol emissions from well outside that region's boundaries. This framing is additionally difficult to apply in decision-making contexts, as mitigation targets are generally set in terms of emissions rather than concentrations or radiative forcing. Several studies have looked at the effects of all aerosols within a given latitude band[19–23], but this is not a geographic delineation along which mitigation decisions, emissions accounting, or conjunct trends in aerosol emissions occur. Other studies have assessed the relative effects of historical emissions or scalings thereon in different countries[17,23–26]. However, these emissions are unequal and therefore cannot be used straightforwardly to create a quantitative framework for assessing the relative importance of aerosol emissions from different regions in the context of the climate metrics used in many scientific and policy frameworks.

Here, we address a fundamental outstanding question underpinning both the consideration of aerosol emissions in climate policy and the analysis of their evolving role in global and regional climate change and climate sensitivity: can the temperature effects of a unit of aerosols emitted from any major emitting economy be considered equivalent? We analyze the relative climate effects of identical combined sulfate, black carbon, and organic carbon aerosol emissions—equivalent to China's total annual year 2000 anthropogenic emissions[1]—sourced from eight major past, present, or projected future emitting regions (Fig. 1a) in a global atmospheric general circulation model coupled to a mixed-layer ocean model (see Methods). These three aerosol species drive the vast majority of anthropogenic aerosol forcing over the historical period[27]. They have historically varied in tandem at the national-scale, as economy-wide transitions have driven aerosol emissions changes over the historical period[1], and are projected to continue to do so in future[28,29]. This novel design provides a crucial advancement by allowing for a unique one-to-one comparison (cleanly interpretable in scientific and decision-making contexts) between aerosol emissions (the perturbation unit that most directly corresponds to mitigation decision-making) in several past, present, or projected future major emitting geopolitical regions (approaching the geographic delineations along which mitigation decisions and emissions accounting are conducted). Our results demonstrate substantial inequalities in both the magnitude and spatial distribution of temperature effects of aerosol emissions located in different major emitting economies. These divergent temperature effects are fundamentally driven by differences in the degree to which each region's emissions and the resulting distribution of radiative effects generate remote feedback processes, and reveal new challenges for understanding and addressing the global and regional climate influence of aerosols.

## Results

**Divergent magnitude and distribution of temperature effects**. Our simulations reveal substantial inequality in the global-mean temperature response to identical total annual aerosol emissions in each region (Fig. 1b, $y$-axis). The largest global mean temperature change ($-0.29 \pm 0.01\,°C$, induced in our simulations by emissions from Western Europe) is approximately 14 times larger than the smallest global mean temperature changes ($-0.02 \pm 0.01\,°C$ induced by emissions from India). Global-mean temperature effects correlate roughly with latitude, with higher latitude emitting regions typically generating stronger temperature effects than lower latitude regions, a behavior also seen in previous studies of the response to aerosol forcing in different latitude bands[19]. However, substantial differentiation occurs within latitude bands. For example, emissions sourced from East Africa generate

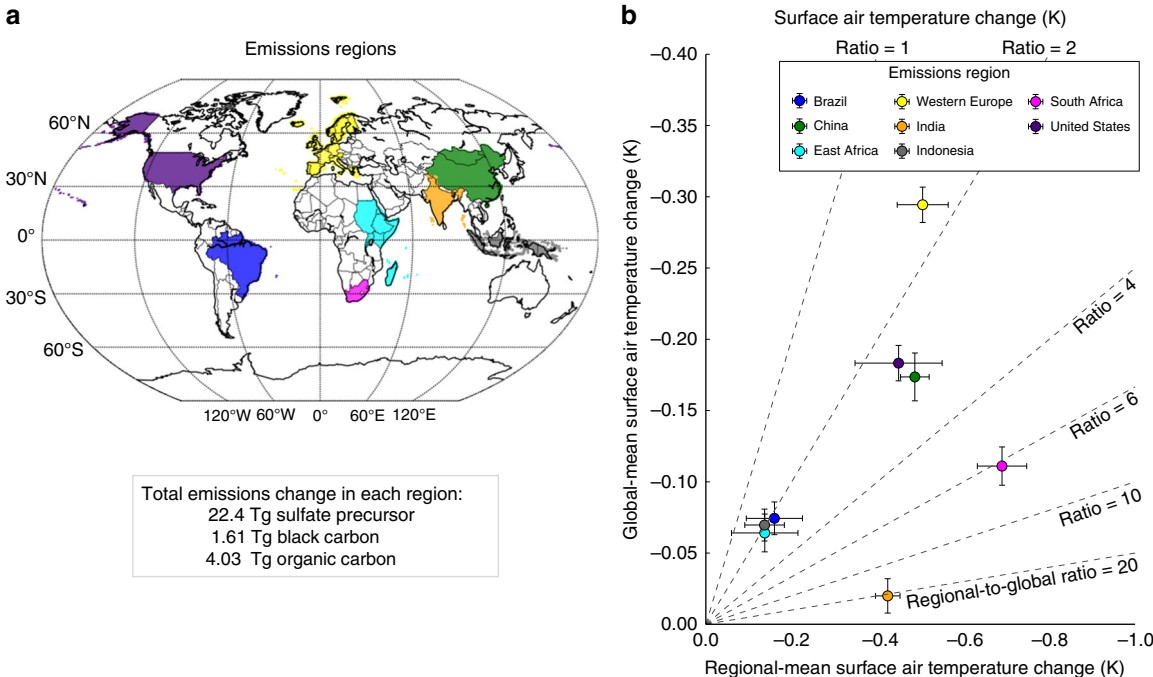

**Fig. 1** Global and regional-mean temperature responses to identical aerosol emissions in different regions. Identical total annual emissions of sulfate and black carbon aerosol from eight major past, present, and potential future emitting regions (**a**) result in global-mean cooling spanning a 14x range (**b**, *y*-axis), differing substantially in the degree to which that cooling is felt in the emitting region (**b**, *x*-axis). Diagonal lines (**b**) indicate the ratio of regional- to global-mean cooling. Error bars in **b** capture the standard error

approximately four times more cooling ($-0.06 \pm 0.01\,°C$) than emissions from India ($-0.02 \pm 0.02\,°C$) at similar latitudes—demonstrating the importance of our region-by-region focus.

There are also important regional distinctions in the degree to which the temperature changes due to aerosol emissions are likely to be concentrated in the emitting region versus being felt globally (Fig. 1b, placement relative to diagonals). Because aerosols generally remain concentrated near their source and most strongly influence surface temperature by attenuating incoming solar radiation to the regions they overlie, they cool the emitting region more strongly than the global mean in all cases (Figs. 1b, 2). However, although Indian emissions produce the smallest global-mean cooling effects, their impacts are much more strongly concentrated within the emitting region than for other regions. India experiences cooling from its own emissions at a level ~21 times greater than the global mean ($-0.42 \pm 0.03\,K$ versus $-0.02 \pm 0.01\,K$), while Western European, and Indonesian emissions generate localized cooling ($-0.50 \pm 0.06$, and $-0.14 \pm 0.05\,K$, respectively) that is less than two times as great as the respective global-mean cooling ($-0.29 \pm 0.01$, and $-0.07 \pm 0.01\,K$, respectively). As a result, even though Western European emissions produce ~14 times the global mean cooling effect of Indian emissions, India and Western Europe experience comparable regional-mean cooling from their own emissions. In other words, regions like Western Europe, Indonesia, and (to a slightly lesser extent) the United States strongly export the climate impacts of their emissions, while regions like India more strongly experience the cooling effects of their own emissions.

**Role of of aerosol burden and radiative forcing differences**. We find that differences in global-mean temperature response resulting from identical emissions emerge at each of several steps between initial emission and eventual temperature effect. Aerosol emissions impact temperature when their resulting atmospheric

concentrations interact with atmospheric radiation, both directly and through aerosols' rapid effects on cloud radiative properties and amount. These radiative interactions then modify the top-of-atmosphere energy balance, which creates a global-mean temperature change mediated by a series of feedback processes. Differences in the temperature change induced by each regional emission can emerge at each step in this process: from emission to atmospheric burden, from burden to top-of-atmosphere radiative forcing, and from top-of-atmosphere radiative forcing to temperature change.

The total atmospheric aerosol burdens generated by the identical regional emissions are spread between 128–233 Gg of sulfate aerosol, 9.29–15.7 Gg of black carbon aerosol, and 23.1–54.2 Gg of organic carbon aerosol (Supplementary Table 1; spatial distributions shown in Supplementary Figs. 1, 2, and 3, respectively). However, the atmospheric burdens of the individual aerosol species generated by each regional emission are largely uncorrelated with the regional emissions' relative potency at changing global-mean temperature (Supplementary Fig. 4). Thus, the disparity in temperature effects does not arise solely from a disparity in the atmospheric lifetime and resulting atmospheric burden of aerosols emitted from each region, but also through the differential generation of climate forcing and feedbacks by those burdens.

How do the aerosol burdens from each region's emissions translate into radiative forcing, which in turn drives the global mean temperature change? The radiative effects of sulfate (a global-mean cooling agent), black carbon (a global-mean warming agent), and organic carbon (a global-mean cooling agent, though with minor shortwave absorbing properties) will counteract each other in driving the radiative forcing. This cancellation can be accommodated by using an aggregate measure of aerosol optical properties, such as aerosol optical depth. In order to capture the cancellation between the absorbing and scattering aerosol burdens, we calculate this as the ratio between

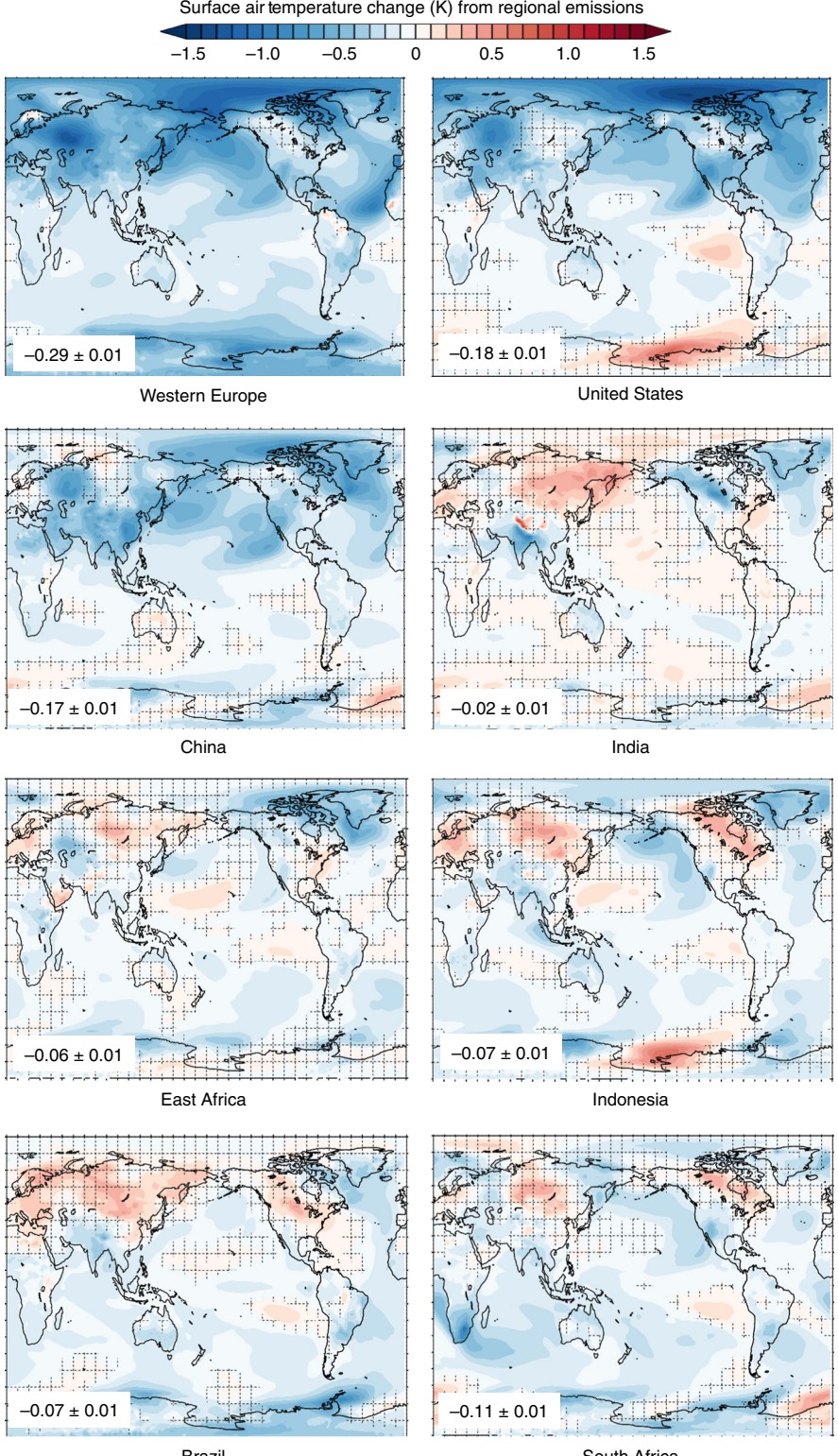

**Fig. 2** Patterns of surface air temperature response to identical aerosol emissions from eight regions. Spatial patterns of surface air temperature change in response to identical aerosol emission in each the eight regions are shown. Global mean temperature change (K) with standard error is shown in bottom left. Grid markings indicate regions that are not statistically significant at the 95% confidence level via *t*-test

the change in global-mean total aerosol optical depth and global-mean absorbing aerosol optical depth caused by each regional emission.

The ratio of total to absorbing aerosol optical depth explains approximately 60% of the variance ($R = 0.60$, $0.44$–$0.69$) in

global-mean top-of-atmosphere effective radiative forcing (ERF) from each regional emission (Fig. 3a). We calculate the ERF as the top-of-atmosphere radiative imbalance induced by the regional emission after the atmosphere and land surface has been allowed to respond (see Methods). The variance in ERF is thus largely

explained by the rate of global-mean cancellation between the absorbing and scattering aerosol burdens resulting from each regional emission, arising from differences in aerosol mixing rates, relative altitudes, and other microphysical and radiative factors[30,31]. The remaining variance may be explained by (1) the radiative environment in which the aerosol population is interacting, created by regional differences in climatological cloud cover or background aerosol; or (2) the particular cloud/convective environment in which the aerosols are placed and the influence this has on the manifestation of the aerosols' indirect and semidirect effects on clouds[32,33]. We find that the ratio of total aerosol optical depth to absorbing aerosol optical depth is approximately as well correlated with clear-sky ERF ($R = 0.58$, 0.31–0.61) as it is with the total ERF, indicating that the presence or absence of climatological cloud does not substantially impact the translation of the atmospheric aerosol burdens into radiative forcing.

**Drastic inequalities in forcing efficacy.** Although the rate of global-mean cancellation between the absorbing and scattering aerosol burdens is somewhat correlated with the top-of-atmosphere radiative forcing generated by each emitting region, the relative top-of-atmosphere effective radiative forcings are not well correlated with the relative global mean temperature change (Fig. 3b), indicating a substantial divergence in forcing efficacy depending on emitting region. The forcing efficacy of each regional emission—i.e., the global-mean temperature change per unit top-of-atmosphere effective radiative forcing (captured in Fig. 4a and by placement relative to the diagonal lines in Fig. 3b)—range from $0.24 \pm 0.14\,\mathrm{K(Wm^{-2})^{-1}}$ to $1.3 \pm 0.06\,\mathrm{K(Wm^{-2})^{-1}}$. This constitutes a factor of five range in forcing efficacy between emitting regions, roughly twice the relative range in estimated forcing efficacy between global historical aerosols and greenhouse gases[7,34]. Emissions from the U.S. and Western Europe have the largest forcing efficacies ($1.32 \pm 0.06$ and $1.09 \pm 0.07\,\mathrm{K(Wm^{-2})^{-1}}$, respectively), producing outsized temperature responses for the effective radiative forcing they generate. Emissions from regions like Brazil, meanwhile, produce a comparable effective radiative forcing to emissions from Western Europe or the U.S., but generate much smaller global-mean temperature change (Fig. 3b). These efficacy differences highlight the shortcomings of using global-mean radiative forcing to estimate the climate effects of highly regionalized forcings.

**Differential excitement of feedbacks.** Differing forcing efficacy is fundamentally driven by differences in how the particular spatiotemporal distribution of global- and annual-mean top-of-atmosphere radiative forcing excites feedback processes that contribute to eventual temperature change. This forcing-feedback framework can be represented mathematically as:

$$dF = \frac{\partial R}{\partial T}dT + \sum_i \frac{\partial R}{\partial X_i}\frac{\partial X_i}{\partial T}dT$$

The climate system balances an initial radiative forcing ($dF$) through the radiative effects of a change in temperature ($\frac{\partial R}{\partial T}dT$), which is either amplified or damped by the top-of-atmosphere radiative effects of temperature-sensitive changes in factors like surface albedo, clouds, and water vapor ($\sum_i \frac{\partial R}{\partial X_i}\frac{\partial X_i}{\partial T}dT$). The degree to which a given initial radiative forcing excites these feedbacks will determine the extent to which the temperature must change to achieve re-equilibration. The differing spatial distributions of the radiative forcing (Supplementary Fig. 5) versus the surface temperature responses (Fig. 2) demonstrates a role for these remote feedback processes in setting the temperature responses to each regional emission.

We find that the differences in forcing efficacy across emitting region can be largely explained by differences in the degree to which the top-of-atmosphere radiative forcing from each regional emission excites top-of-atmosphere surface albedo radiative feedbacks and cloud radiative feedbacks. For each feedback process, we characterize its relative rate of excitement by a given region's emissions as the radiative gain: the ratio of radiative perturbation from the feedback ($\frac{\partial R}{\partial X_i}\frac{\partial X_i}{\partial T}dT$) to the initial radiative forcing generated by the emission ($dF$). Water vapor feedbacks show relatively small and uncorrelated differences in radiative gain across emitting region (Supplementary Fig. 6). However, surface albedo feedbacks—driven primarily by sea ice changes—and cloud feedbacks vary in correspondence with the differences in efficacy (Fig. 4 and Supplementary Fig. 6). The regional differences in the combined radiative gain from the cloud and surface albedo feedbacks (Fig. 4b) explains 84% ($R = 0.84$, 0.71–0.91) of the variance in efficacy across regional emissions (Fig. 4a).

The differences in radiative gain from surface albedo feedbacks and the associated forcing efficacies sort roughly by latitude of emissions. The surface albedo feedback is dominantly driven by sea ice changes (Supplementary Fig. 5) in both the Arctic and Antarctic, and manifests in increased spatial extent and temporal duration of the sea ice. Forcing from Western European and Chinese emissions strongly increases sea ice in both the Arctic and Antarctic, inducing a strong surface albedo feedback radiative gain, while forcing induced by Indian and Brazilian emissions has relatively little effect. The sea ice responses are not strongly spatially collocated with the respective radiative forcings (Supplementary Fig. 5) or atmospheric aerosol burdens (Supplementary Figs. 1, 2, and 3), indicating that the polar effects primarily result from changes in atmospheric circulations that control the energy balance of and/or sea ice dynamics in the polar regions, rather than in situ forcing or aerosol deposition onto the ice. This aligns well with previous studies that show strong dependence of sea-ice albedo feedbacks on the meridional placement of forcings[34,35].

The inter-regional differences in radiative gain associated with cloud feedbacks also help to explain the inter-regional differences in forcing efficacy, and are partially driven by the climatological cloud environment with which each region's aerosol emissions interact. Aerosols emitted in India, Indonesia, and Brazil produce large localized cloud changes (Supplementary Fig. 7), associated with the dynamical and thermodynamical effects of the localized aerosol forcing acting on the strongly convective cloud environment in these regions[36–39]. The cloud changes in response to all regional emissions are primarily dynamical or thermodynamical—rather than microphysical—in nature, as aerosol indirect effects (captured by the change in cloud droplet number concentration, Supplementary Fig. 7) are locally confined and relatively uncorrelated with the maxima in cloud change. India's strong cloud feedback gain and weak surface albedo feedback gain counteract each other in setting the relative overall efficacy of the radiative forcing from Indian emissions.

In our simulations, the cloud feedbacks generated in several regions largely manifest through a north–south shift in tropical cloud cover (Supplementary Fig. 7) associated with the intertropical convergence zone (ITCZ). This meridional ITCZ shift results from the large-scale atmospheric circulation adjusting to compensate for the hemispheric radiative imbalance induced by the localized aerosol forcing and its climate effects[40–43]. This is likely amplified by the surface albedo feedback to each regional emission, which will generate its own hemispheric energy

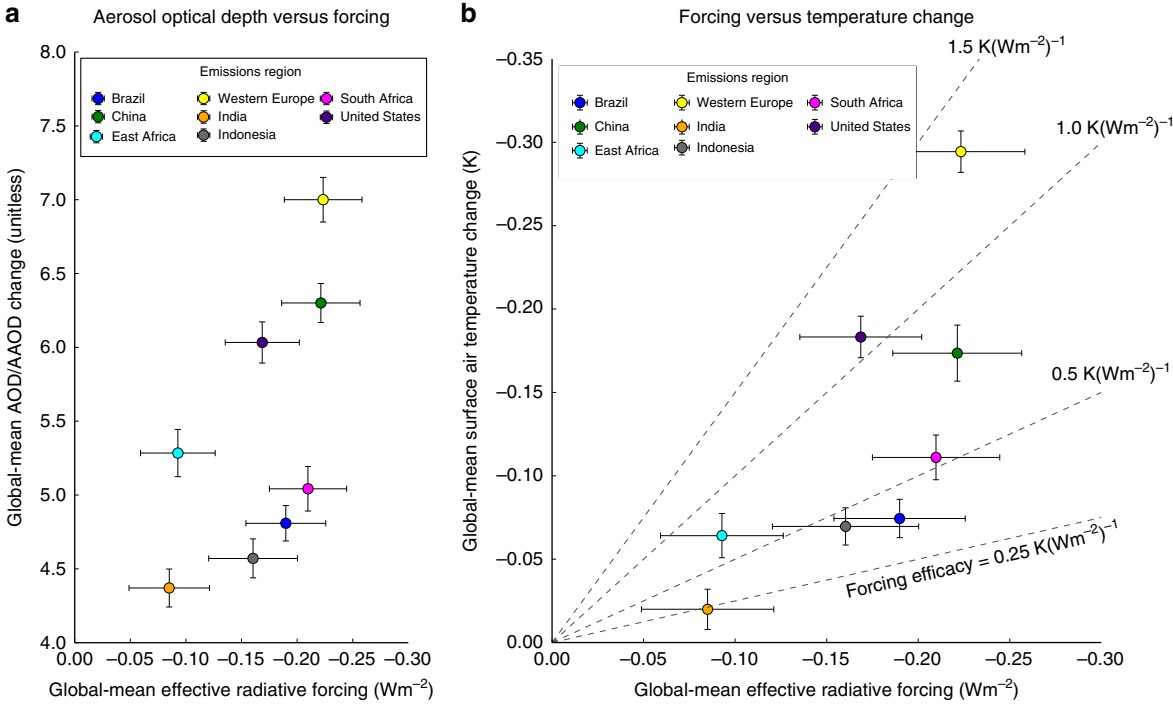

**Fig. 3** Relationships between aerosol burden, radiative forcing, and temperature change. The ratio of aerosol optical depth produced by the atmospheric burden of sulfate, black carbon, and organic carbon to its absorbing component (AOD/AAOD) (**a**, *y*-axis) explains 60% of variance in effective radiative forcing (**a**, **b**, *x*-axis), which in turn has differing efficacy (**b**, diagonals) at producing global mean temperature changes (**b**, *y*-axis). Error bars capture the standard error

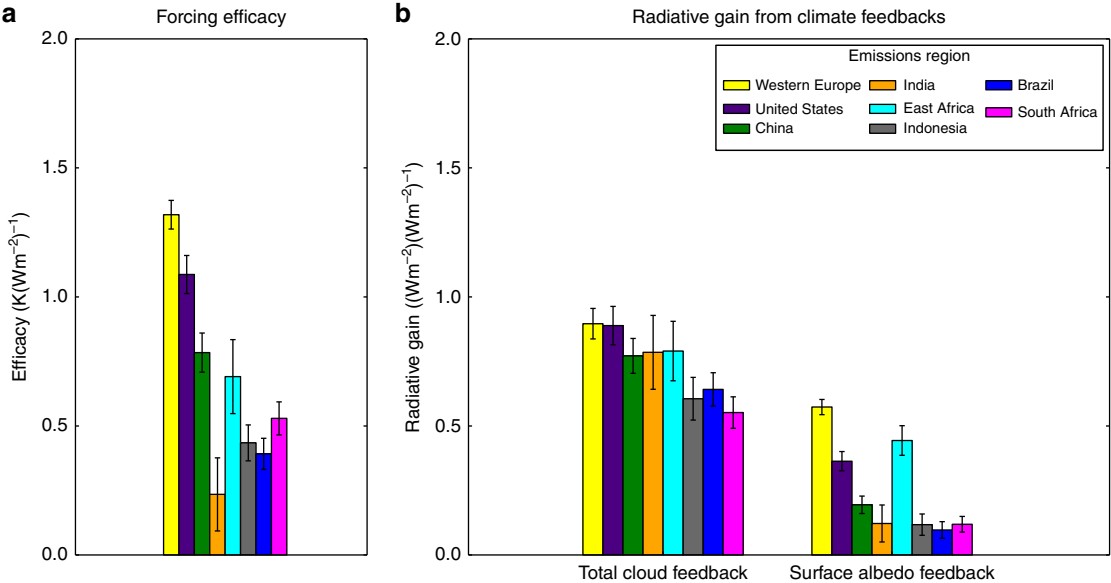

**Fig. 4** Radiative feedbacks versus forcing efficacy. Differences in forcing efficacy (**a**) are largely explained by variance in the global-mean radiative gain from surface albedo and cloud feedbacks (**b**)—i.e., the additional top-of-atmosphere flux change from the feedback due to a unit of effective radiative forcing—across emissions regions. Error bars capture the standard error

imbalance[44]. Even in the presence of Southern Hemispheric emissions, Arctic sea ice increases more strongly than Antarctic sea ice in all cases (Supplementary Fig. 8). This is likely attributable to the stronger overall regional climate sensitivity of the Arctic relative to the Antarctic[45–47]. This common hemispheric imbalance in sea ice response contributes to the comparable ITCZ shifts seen in response to many of the regional aerosol emissions.

## Discussion
These results demonstrate that geographic location substantially influences the cooling potential of a given aerosol emission. Crucially, countries that historically have or presently do account for the majority of anthropogenic aerosol emissions—Europe, the U.S., and China—are the regions whose emissions have the largest cooling potential. Meanwhile, regions with rapid current emissions growth and/or that are projected to be the dominant

sources of anthropogenic emissions going forward—such as India and East Africa—have the smallest cooling potential. This suggests that the historical spatial distribution of anthropogenic emissions may have had a larger cooling potential than will the emissions distribution of the future. Aerosol emissions are projected to decline over the 21st century, as countries increasingly value the air quality benefits of aerosol mitigation[28,48]. However, these findings suggest that the rate at which anthropogenic aerosol emissions offset global-mean greenhouse gas-driven warming may decline more rapidly than changes in global total emissions alone would indicate, due to the additional effects of the simultaneous spatial redistribution of emissions.

While fully coupled climate models will capture the effects of these changes in overall rate of offsetting, the reduced form integrated assessment models (IAMs) and earth system models of intermediate complexity (EMICs) used in many impact assessment settings would fail to capture this signal. The widely used dynamic integrated climate-economy (DICE) integrated assessment model, for example, assumes a time-constant translation of aerosol emissions into temperature effects in calculating future global-mean climate change[18]. It would therefore overestimate the future offsetting effect of anthropogenic aerosol emissions, if their global distribution evolves toward lower potency regions as our results suggest.

These findings have additional implications for estimates of climate sensitivity from historical observations. Estimates of the transient climate response (TCR) and equilibrium climate sensitivity (ECS) to a doubling of $CO_2$ have been attempted from global-mean historical temperature and forcing estimates[11,13]. Recent studies have argued that the different efficacies of individual forcings operating over the historical period must be factored into these observationally-based estimates of TCR and ECS, but do so using time-invariant estimates of efficacy[11,13]. The TCR and ECS are estimated via a statistical fit through the distribution of temperature change versus forcing over historical observations, using the same efficacy across the time series. Results from this study suggest that the efficacy of global aerosol forcing may have evolved substantially over the historical period as different regional emissions dominated the global total emissions. This would suggest the need for a time-evolving value for overall aerosol forcing efficacy over the observational record, potentially resulting in different estimates of TCR and ECS from these methods.

Finally, the inequality in the climate impacts of aerosols from different emitting regions has substantial ramifications for climate accounting and aerosol mitigation policy. These findings highlight the importance of recognizing differences in relative climate influence at the scale of regional aerosol emissions, as aerosols are increasingly factored into international climate policy discussions. Studies to date on the climate influence of aerosols' heterogeneity have primarily focused on the total global distribution or on atmospheric concentrations or radiative forcings within certain latitude bands[6,7,19]. The decisions influencing aerosols' climate effects, however, manifest at the emissions level along political boundaries; increases are generally driven by national or multinational economic policies that encourage emissions-intensive economic activity, while decreases have been largely driven by regional air quality concerns[48]. A mismatch, therefore, has existed until now between the regional scale and surface emissions framing of aerosol-relevant decision-making and the hemispheric scales and atmospheric focus of previous scientific analysis.

Although aerosol emissions mitigation (or its opposite) has historically occurred outside the purview of climate policy, efforts are emerging to factor the climate damage from warming revealed by aerosol mitigation into cost-benefit analyses that have historically considered only air quality benefits. Our findings highlight that calculation of these climate damages must be regionally-specific, both in determining global-mean penalties

from different emitting regions and in evaluating the regional distribution of those penalties. The pronounced divergence in forcing efficacies seen here for national-scale emissions changes—the scale at which climate accounting schemes and mitigation cost-benefit analyses will be undertaken—highlights the danger associated with using global-mean radiative forcing and metrics based on it, such as global warming potentials, as a universal conversion factor for aerosols' global-mean climate impacts across different emitting regions. These results also suggest that for certain regions, such as India, climate impacts may be dramatically stronger in the mitigating region than outside it, while for others it may not. Such inter-regional differences will influence the rate at which the localized climate penalty from revealed warming counteracts the localized air quality benefits of the aerosol mitigation.

These factors demonstrate that evaluation of the social cost of climate impacts from aerosol mitigation will need to be more regionalized than for long-lived greenhouse gases. The particulars of the magnitude and spatial distribution of each regional emission's temperature effects may differ depending on the climate model used, given the substantial spread that exists in model treatment of aerosol processes[49–51]. However, our results demonstrate that major inter-regional differences do emerge, with substantial scientific and policy implications, providing an important first step in motivating future multi-model assessment using the regional emissions framework laid out here.

## Methods

**Model.** Simulations for this study were conducted in the National Center for Atmospheric Research Community Atmosphere Model 5 (NCAR CAM5), the atmospheric component of the Community Earth System Model 1[52], coupled to a mixed-layer ocean. Mixed-layer coupling provides benefits in decreased computational intensity compared to full-ocean coupling, and has been found to lead to similar responses to full ocean coupling in the CESM model suite using an earlier version of the CESM atmospheric model[53]. This similarity is expected to hold with use of CAM5[54]. Simulations using CAM5 coupled to a slab ocean have been widely used in the peer-reviewed literature[54–56], including to assess the climate response to aerosols[57,58]. NCAR CAM5 contains a fully interactive aerosol scheme, which transports and removes the emitted aerosols according to the model's meteorology. We use the CAM5 model with its three mode modal aerosol module (MAM3)[59], containing internal mixing of black carbon, sulfate, and organic carbon with other aerosol species using the volume mixing rule. Refractive indices for sulfate and organic carbon are taken from Hess et al.[60] and for black carbon is taken from Bond et al.[61]. CAM5 also includes aerosol indirect effects on clouds and the radiative effects of black carbon deposition on ice.

**Simulations and analysis.** Nine 100-year, repeating annual cycle simulations were conducted in CAM5 coupled to the mixed-layer ocean: one control simulation, and eight regionally perturbed simulations. The control simulation is a year 2000 climate with non-biomass burning anthropogenic black carbon, organic carbon, sulfur dioxide ($SO_2$), and sulfate ($SO_4$) emissions fields set to 1850 values. In each of the eight regionally perturbed simulations, the relevant region is populated with that region's year 2000 values, scaled at every regional grid point and time step to achieve additional total annual emissions equivalent to China's total year 2000 values: 22.4 Tg sulfate precursor, 1.61 Tg of black carbon emissions, and 4.03 Tg of organic carbon emissions. The 1850 and 2000 baseline emissions fields on which these are based are CAM5's standard historical emissions fields[1], and the resulting emissions fields used to drive simulations are publicly accessible to allow for replication in other model suites (see Data availability).

The eight regions were defined according to the Intergovernmental Panel on Climate Change's regional definitions, and are shown in Fig. 1a. These eight geopolitical regions were selected to sample major past, present, or projected future emitting economies located in a range of climate regimes. Western Europe and the United States, major emitting regions over the historical period, were chosen to sample the response to aerosols emitted in Northern Hemisphere mid-latitude climate regions with different longitudinal locations and associated storm track regimes. India and China, current major emitting regions, were chosen to capture the response to two different monsoonal paradigms. The projected potential future emitting regions[62], Indonesia, Brazil, East Africa, and South Africa, were chosen to assess the impact of aerosols emitted in the following respective climate regimes: the deep convective western Pacific warm pool region, the Pacific and Atlantic basin branches of the Intertropical Convergence Zone, and the Southern Hemisphere midlatitudes. The regional signals cited throughout the manuscript are

calculated as the difference between the corresponding regionally perturbed simulation and the control simulation over the final 60 years of each simulation.

Although black carbon, sulfate, and organic carbon aerosol have somewhat opposing optical properties and global-mean top-of-atmosphere radiative effects, which can complicate analysis when they are co-emitted, we choose to include all of these species in each simulation. The species are co-emitted by many industrial processes, and their mitigation and growth often occur in tandem—a characteristic seen in their Representative Concentration Pathway and Shared Socioeconomic Pathway trajectories[28,29]. Analyzing the climate effects of their simultaneous mitigation or growth, therefore, is likely to provide a better proxy for the climate effects of economy-wide transitions in aerosol emissions, as has been the historical norm and as is projected across the current suite of emissions scenarios used by the climate modeling and policy communities. Simulations that include only one aerosol species may not be additive, particularly in regional responses[63], reducing their utility in assessing the climate response to changes in multiple aerosol species simultaneously, as has been the dominant mode of aerosol emission change in reality. Future work will aim to illuminate the degree of additivity in the multi-species response through comparison with a planned suite of complementary single species simulations.

Effective radiative forcing (ERF) values are calculated using a suite of 9 CAM5 simulations formulated in the same way as the mixed-layer ocean simulations described above, but with the mixed-layer ocean and sea ice module replaced with a repeating annual cycle of observed year 2000 sea surface temperatures (SSTs) and sea ice coverage. These simulations are run for 60 years, the model equilibrates within the first 20 years, and the final 40 years of the simulation are used in the ERF calculation. The ERF is calculated as the top-of-atmosphere radiative imbalance between the control and regional fixed-SST simulation, i.e., after atmospheric and land surface temperatures have been allowed to adjust to the regional emissions. This follows the Fixed SST radiative forcing definition described in Myhre et al.[27].

**Feedback calculations**. Top-of-atmosphere radiative feedbacks are calculated via standard radiative kernel methods[64–66], using radiative kernels generated in the NCAR CESM-CAM5 model[67] and associated kernel calculation code provided by Pendergrass (https://github.com/apendergrass/cam5-kernels). Use of radiative kernels generated within the same climate model as the simulations to which they are applied, as is the case here, is generally considered to be best practice for reducing errors associated with differences in radiative transfer codes between climate models[64,68].

The radiative kernel for a given feedback process consists of gridded monthly values of the radiative perturbation induced by a prescribed change in a given feedback process, such as surface albedo or atmospheric water vapor. These gridded kernels are then multiplied by the monthly change in the relevant feedback process for each of the regional emissions signals (see Simulations and analysis), normalized by the prescribed change used to derive the kernel. This process is used to calculate the feedbacks due to all process other than the cloud radiative feedback. The cloud radiative feedback is calculated using the adjusted cloud radiative effect method[64]. In this method, a cloud radiative effect is calculated as the difference between the all-sky and clear-sky (i.e., non-cloud-permitting) top-of-atmosphere radiative imbalance. This cloud radiative effect is then adjusted for potential contamination by non-cloud factors through subtraction of the kernel-calculated cloudy-sky (i.e., all-sky kernel minus clear-sky kernel) values of the other feedbacks. This method has been shown to provide a reliable estimate for cloud feedbacks relative to other, more computationally prohibitive, methods[47].

**Statistical testing**. Error ranges for the values in Figs. 1, 3, and 4, Supplementary Figs. 4 and 6, and Supplementary Table 1 are calculated as the standard error of the mean, using the final 60 years of each simulation's data, with the exception of the ERF values in Supplementary Table 1, which use the final 40 years of the Fixed SST simulations' data. In each case an effective sample size is calculated, accounting for any autocorrelation in the time series[69]. The statistical significance masking in Fig. 2 and Supplementary Figs. 1, 2, 3, 7, and 8 is calculated using a one-sample t-test, again using the final 60 years of model data with an effective sample size adjusted for autocorrelation[69]. Supplementary Fig. 5 is calculated using the final 40 years of the Fixed SST simulation data. Regions with gridlines in these figures indicate signals that are not significant at the 95% confidence level. Error ranges provided for R values are calculated using jackknife resampling[70], in which the R value is recalculated seven times, dropping one of the regions each time.

**Code availability**. The code for the NCAR CESM model is publicly available at http://www.cesm.ucar.edu/models/. The code used to set up the simulations used in this study and to analyze data is available from the authors upon request.

**Data availability**. Emissions input fields used to drive the simulations are available at https://drive.google.com/drive/folders/1ggEJ4J6vrwk6HDYAFI2lgYhmYjRkJ8kk?usp=sharing. Simulation output evaluated in this study are available from the authors upon request.

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

## Acknowledgements

The authors wish to thank Anna Possner and Patrick Brown for helpful comments on early drafts of this work. This work was partially supported by National Science Foundation Grant NSF CNH-L 1715557.

## Author contributions

G.G. Persad conceived of the study, ran model simulations, conducted analysis, and wrote the manuscript. K.C. provided input on data analysis and simulation set-up, and contributed to editing of the manuscript.

## Additional information

**Competing interests:** The authors declare no competing interests.

