## [Peer Review File · Nature Communications]

Reviewer #1 (Remarks to the Author):

Review of manuscript from Persad & Caldeira "Divergent Global-Scale Temperature Effects from Identical Aerosols Emitted in Different Regions"

This work estimated the effect on global temperatures of the same aerosol precursors forming sulfate and black carbon emitted over eight distinct regions of the world. The authors do not restrict their analysis to quantifying the change in surface temperature around the globe or the effective radiative forcing but they analyse the relative roles of changes in shortwave, longwave radiation, water vapor, surface albedo and temperature on the overall radiative change.

This study is well led, the results could be far-reaching as they clearly indicate that the world regions that have been the strongest emitters as of today are much more sensitive in terms of temperature change than emerging regions where most of the aerosol will be emitted in the later part of the XXIst century.

The statistics and the methodology are clearly described and I have two comments that are one major and the second one minor.

The first one is clearly identified by the authors in the last sentence of the article and has to do with the model used NCAR CAM5 coupled to a slab ocean. NCAR Research Community Atmosphere Model 5 (CAM5) has been documented to have clouds that respond in a very sensitive way to aerosols. Hence, there is no warranty that other models would quantitatively show the same results. The authors allude to this on lines 256 to 260:

" While the particular magnitude and spatial distribution of each regional emission's temperature effects may differ depending on the climate model used – a fact that encourages future multi-model assessment of the impact of emission location – our results demonstrate that major inter-regional differences do emerge, with substantial scientific and policy implications."

My second concern is minor and can be taken into account by the authors.

Since other coupled models should replicate these experiments to revisit quantitatively their significance, the authors need to give much more information in the way the aerosols are treated and not refer to other previously written papers for this information. At a minimum, they need to specify the mixing rule that is used since they treat the aerosol as internally mixed, and, they should give the refractive indices for sulphate and black carbon that they use so that other groups can do comparable experiments.

My overall judgment is that these results are sufficiently new and far reaching that they can be published in Nature Communications.

I have 2 more suggestions for the authors.

1/ the eight different regions have very different annual mean cloud cover. This could influence the sensitivity of the response to the same emissions in each region. How did you analyse this aspect?

2/ The response in temperature to aerosols is easier to analyse when you link it to top-of-atmosphere radiative perturbation (see Miller et al., 2014 for instance). How/why did you choose to analyse the surface radiative forcing?

Yves Balkanski

Reference

Miller, R.L., P. Knippertz, C. Pérez García-Pando, J.P. Perlwitz, and I. Tegen, 2014: Impact of dust radiative forcing upon climate. In *Mineral Dust: A Key Player in the Earth System*. P. Knippertz, and J.-B.W. Stuut, Eds. Springer, 327-357, doi:10.1007/978-94-017-8978-3_13.

Reviewer #2 (Remarks to the Author):

The study by Persad and Caldeira investigates climate responses to identical amounts of aerosol emissions from various key regions of the globe. The authors highlight the divergent responses found in their model for the different emission regions, and discuss the fact that the influences roughly increase with emission latitude. They also analyze the reasons behind this divergence, and in particular they explore the differences of forcing efficacies between the regions using the forcing-

feedback framework to conclude that sea ice and cloud feedbacks are largely responsible. Finally, implications for policy are outlined.

The study is well written, the topic is important, and the experiments are interesting. The discussion of implications for policy is also nice. However, I have some reservations when it comes to the suitability of the article for this particular journal, as I find the novelty of the study/findings to be somewhat overstated by the authors. I also have some concerns when it comes to the methodology followed. More specifically:

a) A key feature/finding, i.e. the stronger responses to forcing in extratropical versus tropical locations, is not discussed for the first time in the scientific literature. Shindell and Faluvegi (2009) had demonstrated and highlighted the fact that the location of aerosol and other forcings matters for global/regional temperature response in a coupled model (and did similarly for precipitation, in a subsequent paper in 2012), while there were earlier such indications as well in papers such as Forster et al. (2000). This past work which is highly relevant (especially the 2009 paper) is not mentioned in the introductory parts of the current manuscript, where existing work is discussed, but only briefly in the results section (L59). Breaking down the influence by region of emission has also been studied before (Aamaas et al. (2017), not cited in the manuscript; Collins et al. (2013), already cited); it may have been done using a two-step process (i.e. not with a single modeling framework), but these studies provided useful new insights, such as the fact that the influences may also vary with emission region within the same latitude band (e.g. European aerosols were found to be more important for global climate than East Asian aerosols in both studies, as is also found here). What is new in this and some other efforts that are currently ongoing in the community globally is the effort to separate effects from different regions in a seamless, atmosphere-ocean modeling approach (though see comment below regarding the use of a mixed-layer ocean). However, the above literature should have been discussed in the introduction, as currently the novelty of the study appears more pronounced than it actually is.

b) It is not convincing that applying perturbations to two aerosol species (sulfate and black carbon) that have such different optical properties and TOA forcing is a good idea. These species may often be co-emitted, though there are many exceptions (to the extent that they are not exceptions). In fact, several of the most common BC measures that are being suggested for assisting climate change mitigation (for example those discussed in the Integrated Assessment for Black Carbon and Tropospheric Ozone by UNEP – see Box 5.1) do not involve large quantities of sulfur emissions. I would argue that having the efficacies (or temperature potentials) for different pollutants separately is a “cleaner” approach. If one preferred to design an alternative approach that has even more relevance to policy, the most suitable way to go would be to perturb emissions from different sectors in different regions, i.e. all the species emitted by separate sectors, and accounting for the “real” emission factors of different species in different sectors. Finally, I find the last sentence of the justification of this choice (i.e. L296-298) hard to understand.

c) It is not clear to me whether the way that statistical significance of the results is assessed is appropriate in this case. It may be a method sometimes used in such analysis, but I do not believe it is entirely right. The authors show results as 60-year means, as I understand. What would then matter is how much the climate system varies on that timescale, i.e. 60 years. There are modes of variability that occur on multi-decadal timescales, and for those to be captured, one would need to have a picture of how the system (i.e. global climate in this model) could evolve on that timescale in an unforced situation. This is typically done by other performing multiple (or at least a few) ensemble members, or by extending (at least) the control simulation to a multiple of 60 years and then calculating the error from 60-year segments of that run. I am therefore wondering whether the method is adequate, and specifically whether uncertainty has been underestimated? For example, I was quite surprised to see such high confidence in cloud responses to aerosol forcings (even very remotely) in Fig. S6.

d) Mixed-layer ocean model versions are certainly helpful as they involve shorter simulation times, and they often help isolate different oceanic influences (when compared to a full-depth ocean model), but in this case it would have been preferable to use a full-depth ocean model, as the full climate responses in a new equilibrium will have likely been modulated by ocean circulation changes - especially the way that heat is being redistributed across and between the hemispheres. The authors mention that "Responses in CAM5 coupled to a mixed-layer ocean have been shown to be well correlated with responses in CAM5 coupled to a fully dynamical ocean", and cite Bitz et al. (2012). However, Bitz et al. seem to have evaluated CAM4 and not CAM5 which was used here, so the above statement is a bit misleading. Furthermore, I could not locate where the cited paper suggests this correlation (although it is a long and detailed paper and I may have missed this).

Other Comments:

L39-40: Any reason why other types of anthropogenic aerosols were excluded? Similarly (not referring to the same lines), why is only East Africa studied and not West Africa? And why was Eastern Europe not included, given that its emissions are a substantial fraction of European emissions?

Figure 1: Axes labels could be magnified.

Discussion of Figure 2: Is there any dependence of the forcings on the background aerosols existing in different regions? That is worth some mention.

Figure 2, right panel: “2” should be raised to a superscript.

L106: Why not also actually show the relationship between burden and forcing in a similar scatter plot?

L108: This is referring to global imbalance, I presume? Please clarify.

L111: From Fig. 2, the difference seems to mainly be large for sulfate, not for BC.

L111-118: This discussion a bit rushed and hand-wavy. It is important to know which factor(s) may be more important here, at least for the example that the authors have picked (East African vs South African emissions).

Figure 3: Suggest adding “aerosol” before “emissions”.

L158-170: The surface albedo feedback influence seems to be given priority in the discussion (also in the abstract, in terms of order of mention), but the cloud feedback is more dominant (both its mean magnitude but also its absolute difference between e.g. the E. Africa and US or W. Europe simulations, based on Fig. 4).

L190-197: Why would the Brazilian perturbation generate a similar ITCZ shift to the northern mid-latitude perturbations, when its forcing is mainly confined to the Southern Hemisphere? Similarly, why would E. African emissions lead to a similar response, e.g. over the Pacific, even though the emissions are primarily over the Equator. Sea-ice feedbacks on clouds themselves could be an explanation (i.e. hemispherically preferential sea ice feedbacks could affect the inter-hemispheric heating imbalance in a way that favours the specific ITCZ shift pattern), but there are sea-ice changes in the SH too.

L210-213: Yes, useful point to make. But worth noting that climate models inherently include this redistribution of emissions in future scenarios driving them. Therefore this is not a missing process, but possibly it is not a fully appreciated one (e.g. IAMs do not commonly include such regional influences, as the authors mention earlier).

L216: Suggest clarifying that this refers to global temperature change and radiative forcing.

L247-248: This seems somewhat contradictory to the picture described earlier in L13-24. Efficacies should surely matter even on a global scale, no? Or maybe I am misunderstanding something, in which case maybe this means that this could be clarified.

L258: See also the more recent study by Kasoar et al. (2016), which actually focused on regional emission impacts and explored causes of model diversity in the responses in detail.

L299-305: How many years have been used for the ERF calculations? Please mention. The maps on Fig. S3 seem particularly noisy.

Aamaas, B., Berntsen, T. K., Fuglestedt, J. S., Shine, K. P., and Collins, W. J. (2017), Regional temperature change potentials for short-lived climate forcers based on radiative forcing from multiple models, *Atmos. Chem. Phys.*, 17, 10795-10809, <https://doi.org/10.5194/acp-17-10795-2017>.

Collins, W. J., Fry, M. M., Yu, H., Fuglestedt, J. S., Shindell, D. T., and West, J. J. (2013), Global and regional temperature-change potentials for near-term climate forcers, *Atmos. Chem. Phys.*, 13, 2471-2485, <https://doi.org/10.5194/acp-13-2471-2013>.

Forster, P. M. d. F., Blackburn, M., Glover, R. & Shine, K. P. (2000), An examination of climate sensitivity for idealised climate change experiments in an intermediate general circulation model, *Clim. Dyn.*, 16, 833–849.

Kasoar, M., Voulgarakis, A., Lamarque, J.-F., Shindell, D. T., Bellouin, N., Collins, W. J., Faluvegi, G., and Tsigaridis, K. (2016), Regional and global temperature response to anthropogenic SO₂ emissions from China in three climate models, *Atmos. Chem. Phys.*, 16, 9785-9804, <https://doi.org/10.5194/acp-16-9785-2016>.

Reviewer #3 (Remarks to the Author):

This paper investigates how global surface air temperature responds differently to aerosol emissions from different geographic regions. Of especial merit is the demonstration how (a) identical emission, when released in different regions, may lead to differences in (b) atmospheric burden, (c) radiative forcing, (d) feedback and (e) temperature response in order. The results have strong implications for assessment of historical forcing and policy making. I recommend publication after the following issues, mostly minor, are properly addressed.

One comment is that the uncertainties that arise in each of the above steps (a-b-c-d-e) need to be better noted in the paper. It would strengthen the paper to particularly evaluate the uncertainty in effective radiative forcing, which explains the majority of the global warming difference ($R^2=0.55$) but, from Fig S3, appears not explained by the burden difference. Hence, a key question is how the different ERFs result from the burdens. For instance, it is noticed that there is much remote effects and strong cloud adjustment in response to Indian emission, which leads to the small warming potential of the emission in this region - a key conclusion of the paper. The question is how much uncertainty is in this result? Note that it has been recognized that model-simulated adjusted forcing can be very uncertain judged from disparity among different models [e.g., Vial et al. 2013].

Line 13-14. Note the greenhouse gas forcing is also heterogeneous [Huang et al. 2016]. This ought to be noted when comparing aerosol and greenhouse gas forcings and their climate responses.

Line 17. What does “projection” mean here?

Line 39. What does “functionally” mean?

Line 320. The feedback results can be validated, e.g., by checking the radiation closure [e.g., see example given by Huang et al. 2017] and/or checking the sensitivity to the kernel dataset used (note multiple sets are available).

Figures. Some fonts in the labels and legends are too small.

References

Huang, Y., X. Tan and Y. Xia, (2016), Inhomogeneous radiative forcing of homogeneous greenhouse gases, *J. Geophysical Res.-Atmosphere*, 121, doi: 10.1002/2015JD024569.

Huang, Y., Y. Xia and X. Tan, (2017), On the pattern of CO₂ radiative forcing and poleward energy transport, *J. Geophys. Res.-Atmosphere*, doi: 10.1002/2017JD027221.

Vial, J., J.-L. Dufresne, and S. Bony, 2013: On the interpretation of inter-model spread in CMIP5 climate sensitivity estimates. *Climate Dyn.*, 41, 3339–3362, doi:10.1007/s00382-013-1725-9.

Reviewer #1 (Remarks to the Author):

We thank the reviewer for their fair and constructive review of our manuscript. We have taken care to address all issues raised by the reviewer, and have added several improvements to the manuscript that enhance ease of reproducibility of our novel simulation set-up in other model suites and provide increased confidence in the credibility of our single model findings; assess the potential role of climatological cloud cover; and clarify our use of top-of-atmosphere radiative analysis.

Review of manuscript from Persad & Caldeira “Divergent Global-Scale Temperature Effects from Identical Aerosols Emitted in Different Regions”

This work estimated the effect on global temperatures of the same aerosol precursors forming sulfate and black carbon emitted over eight distinct regions of the world. The authors do not restrict their analysis to quantifying the change in surface temperature around the globe or the effective radiative forcing but they analyse the relative roles of changes in shortwave, longwave radiation, water vapor, surface albedo and temperature on the overall radiative change.

This study is well led, the results could be far-reaching as they clearly indicate that the world regions that have been the strongest emitters as of today are much more sensitive in terms of temperature change than emerging regions where most of the aerosol will be emitted in the later part of the XXIst century.

The statistics and the methodology are clearly described and I have two comments that are one major and the second one minor.

The first one is clearly identified by the authors in the last sentence of the article and has to do with the model used NCAR CAM5 coupled to a slab ocean. NCAR Research Community Atmosphere Model 5 (CAM5) has been documented to have clouds that respond in a very sensitive way to aerosols. Hence, there is no warranty that other models would quantitatively show the same results. The authors allude to this on lines 256 to 260:

“ While the particular magnitude and spatial distribution of each regional emission’s temperature effects may differ depending on the climate model used – a fact that encourages future multi-model assessment of the impact of emission location – our results demonstrate that major inter-regional differences do emerge, with substantial scientific and policy implications.”

The reviewer is correct that there is likely to be some model dependence associated with any single model study. A key contribution of this manuscript is its novel experiment design, as the reviewer recognizes, which is computationally intensive and has not—to our knowledge—been attempted by any other modeling group. It is therefore not feasible for us to attempt these simulations in other model suites ourselves, due to computational and access limitations. Our hope is that our findings regarding the substantial divergence in temperature response resulting from identical aerosol emissions in different regions will encourage others in the modeling community to replicate these simulations in other model suites to which they have access. A prior step to such a multi-model effort, however, is demonstrating that the phenomenon is significant enough to warrant additional study in other models – a prior that we have established in this work. Although we agree and expect that the precise quantitative results that other models produce may differ from ours, due to the differences in physics between model suites, we expect that our main qualitative finding—that identical aerosol emissions from major economies can result in substantially divergent global-scale temperature responses meriting consideration in scientific and policy analysis—will hold. Given the widespread use of the NCAR CESM suite in the peer-reviewed literature to assess the climate response to aerosols (Hsieh et al., 2013; Lin et al., 2016; Kim et al., 2016; Deng et al., 2016; Xu and Xie, 2015, etc.), we believe that it is an appropriate model for our purposes.

We have taken the following actions in this revision to address this concern:

- In order to more greatly facilitate future replication of this work by other modeling groups we now provide substantial additional details of our model simulation set-up in response to the reviewer’s comment (see below response at L314-370)

- Additionally, we have expanded our discussion of the above framing at L298-304, as follows: in order to more clearly articulate our agreement with the reviewer's observation that there are likely to be quantitative differences when these models are replicated in other model suites: *"The particulars of the magnitude and spatial distribution of each regional emission's temperature effects may differ depending on the climate model used, given the substantial spread that exists in model treatment of aerosol processes⁴⁹⁻⁵¹. However, our results demonstrate that major inter-regional differences do emerge, with substantial scientific and policy implications, providing an important first step in motivating future multi-model assessment using the regional emissions framework laid out here."*
- We additionally note that, in response to comments by Reviewer 2, we have now redone all simulations to include identical organic carbon aerosol emissions, in addition to black carbon and sulfate aerosol. The resulting simulations, presented in the revised manuscript, now better capture the effects of economy-wide aerosol emissions. In the process of setting up the new simulations, we identified minor logic errors in the original simulations affecting the precise equality of the regional emissions, and so do not include the original simulations alongside these revised simulations. However, the fact that the key findings—that the temperature response to identical emissions in different regions differs widely, that forcing efficacy differs by approximately a factor of five, that regions such as India feel the effects of their own emissions much more strongly than other regions, and that emissions from past emitters have stronger cooling potential than emissions from potential future emitters—are all either upheld or amplified in these new simulations gives us increased confidence in the credibility of these findings.

- Lin, Lei, Zhili Wang, Yangyang Xu, and Qiang Fu. "Sensitivity of Precipitation Extremes to Radiative Forcing of Greenhouse Gases and Aerosols." *Geophysical Research Letters* 43, no. 18 (September 28, 2016): 2016GL070869. <https://doi.org/10.1002/2016GL070869>.
- Kim, Minjoong J., Sang-Wook Yeh, and Rokjin J. Park. "Effects of Sulfate Aerosol Forcing on East Asian Summer Monsoon for 1985-2010." *Geophysical Research Letters*, January 1, 2016, 2015GL067124. <https://doi.org/10.1002/2015GL067124>.
- Deng, Jiechun, Haiming Xu, and Leying Zhang. "Nonlinear Effects of Anthropogenic Aerosol and Urban Land Surface Forcing on Spring Climate in Eastern China." *Journal of Geophysical Research: Atmospheres*, April 1, 2016, 2015JD024377. <https://doi.org/10.1002/2015JD024377>.
- Xu, Y., and S.-P. Xie. "Ocean Mediation of Tropospheric Response to Reflecting and Absorbing Aerosols." *Atmos. Chem. Phys.* 15, no. 10 (May 27, 2015): 5827–33. <https://doi.org/10.5194/acp-15-5827-2015>.
- Hsieh, W.-C., W. D. Collins, Y. Liu, J. C. H. Chiang, C.-L. Shie, K. Caldeira, and L. Cao. "Climate Response Due to Carbonaceous Aerosols and Aerosol-Induced SST Effects in NCAR Community Atmospheric Model CAM3.5." *Atmos. Chem. Phys.* 13, no. 15 (August 5, 2013): 7489–7510. <https://doi.org/10.5194/acp-13-7489-2013>.

My second concern is minor and can be taken into account by the authors.

Since other coupled models should replicate these experiments to revisit quantitatively their significance, the authors need to give much more information in the way the aerosols are treated and not refer to other previously written papers for this information. At a minimum, they need to specify the mixing rule that is used since they treat the aerosol as internally mixed, and, they should give the refractive indices for sulphate and black carbon that they use so that other groups can do comparable experiments.

We now provide additional details on the set up of the experiments to facilitate replication in other model suites. We also provide additional details on the formulation of NCAR CAM5 used in these simulations, including the optical properties attributed to the different aerosol species and the mixing rule implemented. Upon publication of the manuscript, we will provide direct access to the emissions fields used to generate the simulations, and have now included the link where this data will be housed in the data availability statement. We have designed the information contained in these sections to

provide all information necessary for other groups to replicate these simulations in other model suites to which they have access.

We do wish to note that there are a range of aerosol formulations used across the major modeling suites and limited observational validation to distinguish which may be more or less physically robust (e.g. Stier et al., 2013). One hope of having these simulations replicated in other models would be to test the robustness of these results across the same experimental design executed in the other models' aerosol formulations, rather than by requiring those models to replicate the aerosol formulation used in CAM5.

We have included this extensive description of experiment design and model treatment at L314-L370 as follows:

*“NCAR CAM5 contains a fully interactive aerosol scheme, which transports and removes the emitted aerosols according to the model’s meteorology. We use the CAM5 model with its three mode modal aerosol module (MAM3)⁵⁹, containing internal mixing of black carbon, sulfate, and organic carbon with other aerosol species using the volume mixing rule. Refractive indices for sulfate and organic carbon are taken from Hess et al. (1988)⁶⁰ and for black carbon is taken from Bond et al. (2006)⁶¹. CAM5 also includes aerosol indirect effects on clouds and the radiative effects of black carbon deposition on ice. **Simulations and Analysis.** Nine 100-year, repeating annual cycle simulations were conducted in CAM5 coupled to the mixed-layer ocean: 1 control simulation, and 8 regionally perturbed simulations. The control simulation is a year 2000 climate with non-biomass burning anthropogenic black carbon, organic carbon, sulfur dioxide (SO₂), and sulfate (SO₄) emissions fields set to 1850 values. In each of the 8 regionally perturbed simulations, the relevant region is populated with that region’s year 2000 values, scaled at every regional grid point and time step to achieve additional total annual emissions equivalent to China’s total year 2000 values: 22.4 Tg sulfate precursor, 1.61 Tg of black carbon emissions, and 4.03 Tg of organic carbon emissions. The 1850 and 2000 baseline emissions fields on which these are based are CAM5’s standard historical emissions fields¹, and the resulting emissions fields used to drive simulations are publicly accessible to allow for replication in other model suites (see Data Availability).*

The 8 regions were defined according to the Intergovernmental Panel on Climate Change’s regional definitions, and are shown in Figure 1a. ... The regional signals cited throughout the manuscript are calculated as the difference between the corresponding regionally perturbed simulation and the control simulation over the final 60 years of each simulation.

*...
Effective radiative forcing (ERF) values are calculated using a suite of 9 CAM5 simulations formulated in the same way as the mixed-layer ocean simulations described above, but with the mixed-layer ocean and sea ice module replaced with a repeating annual cycle of observed year 2000 sea surface temperatures (SSTs) and sea ice coverage. These simulations are run for 60 years, the model equilibrates within the first 20 years, and the final 40 years of the simulation are used in the ERF calculation. The ERF is calculated as the top-of-atmosphere radiative imbalance between the control and regional fixed-SST simulation, i.e. after atmospheric and land surface temperatures have been allowed to adjust to the regional emissions. This follows the “Fixed SST” radiative forcing definition described in Myhre et al., 2013²⁶. “*

Stier, P., N. A. J. Schutgens, N. Bellouin, H. Bian, O. Boucher, M. Chin, S. Ghan, et al. “Host Model Uncertainties in Aerosol Radiative Forcing Estimates: Results from the AeroCom Prescribed Intercomparison Study.” *Atmos. Chem. Phys.* 13, no. 6 (March 20, 2013): 3245–70.
<https://doi.org/10.5194/acp-13-3245-2013>.

My overall judgment is that these results are sufficiently new and far reaching that they can be published in Nature Communications.

We thank the reviewer for their favorable assessment of the impact of our findings. We hope that the above and below additions to the manuscript serve to address the reviewer’s remaining concerns.

I have 2 more suggestions for the authors.

1/ the eight different regions have very different annual mean cloud cover. This could influence the sensitivity of the response to the same emissions in each region. How did you analyse this aspect?

The reviewer is correct that annual mean climatological cloud cover differs substantially between the regions. There are two potential pathways through which we could see this affecting the sensitivity of response, and we now address both aspects in the revised manuscript.

First, climatological cloud cover could potentially influence the response sensitivity by modifying the effective radiative forcing resulting from identical aerosol burdens via cloud masking or aerosol indirect effects. However, we find that our metric of aggregate aerosol burden (the ratio of aerosol optical depth to absorbing aerosol optical depth) is approximately as well correlated with clear-sky effective radiative forcing ($R=0.58$, $0.31-0.61$), i.e. when clouds are turned off in the model's radiative transfer scheme, as it is with the total effective radiative forcing ($R=0.60$, $0.44-0.69$). This indicates that the presence or absence of cloud does not substantially influence the translation of the atmospheric aerosol burdens into radiative forcing.

We include the above discussion on the potential role of climatological cloud cover in the ERF values at L145-153 as follows:

“The remaining variance may be explained by 1) the radiative environment in which the aerosol population is interacting, created by regional differences in climatological cloud cover or background aerosol; or 2) the particular cloud/convective environment in which the aerosols are placed and the influence this has on the manifestation of the aerosols' indirect and semidirect effects on clouds^{31,32}. We find that the ratio of total aerosol optical depth to absorbing aerosol optical depth is approximately as well correlated with clear-sky ERF ($R=0.58$, $0.31-0.61$) as it is with the total ERF, indicating that the presence or absence of climatological cloud does not substantially impact the translation of the atmospheric aerosol burdens into radiative forcing.

Secondly, the differences in regional climatological cloud cover could also influence the response sensitivity by influencing the strength of cloud radiative feedback generated by the radiative forcing from each region's emissions. Indeed, we find that the variation in the strength of the cloud radiative feedback plays a role in explaining the divergence in forcing efficacy. We expect that India, East Africa, and Indonesia's climatological tropical convective environments play a large role in explaining the strength of the cloud radiative feedback that forcing from those regions' emissions generate (See, e.g., decomposition of cloud feedbacks in Supplementary Figure 6 and cloud response in Supplementary Figure 8). We now include explicit discussion of the role of the climatological cloud cover in this section at L210-L215 as follows:

“The inter-regional differences in radiative gain associated with cloud feedbacks also help to explain the inter-regional differences in forcing efficacy, and are partially driven by the climatological cloud environment with which each region's aerosol emissions interact. Aerosols emitted in India, Indonesia, and Brazil produce large localized cloud changes (Supp. Fig. 8), associated with the dynamical and thermodynamical effects of the localized aerosol forcing acting on the strongly convective cloud environment in these regions³⁵⁻³⁸.”

2/ The response in temperature to aerosols is easier to analyse when you link it to top-of-atmosphere radiative perturbation (see Miller et al., 2014 for instance). How/why did you choose to analyse the surface radiative forcing?

All analysis of radiative forcing and radiative feedbacks is conducted on top-of-atmosphere terms, rather than surface radiative terms. We apologize for not making this more explicit in the original manuscript and now specify the top-of-atmosphere formulation throughout the revised manuscript wherever the radiative analyses are discussed.

Yves Balkanski

Reference

Miller, R.L., P. Knippertz, C. Pérez García-Pando, J.P. Perlwitz, and I. Tegen, 2014: Impact of dust radiative forcing upon climate. In *Mineral Dust: A Key Player in the Earth System*. P. Knippertz, and J.-B.W. Stuut, Eds. Springer, 327-357, doi:10.1007/978-94-017-8978-3_13.

Reviewer #2 (Remarks to the Author):

The study by Persad and Caldeira investigates climate responses to identical amounts of aerosol emissions from various key regions of the globe. The authors highlight the divergent responses found in their model for the different emission regions, and discuss the fact that the influences roughly increase with emission latitude. They also analyze the reasons behind this divergence, and in particular they explore the differences of forcing efficacies between the regions using the forcing-feedback framework to conclude that sea ice and cloud feedbacks are largely responsible. Finally, implications for policy are outlined.

The study is well written, the topic is important, and the experiments are interesting. The discussion of implications for policy is also nice. However, I have some reservations when it comes to the suitability of the article for this particular journal, as I find the novelty of the study/findings to be somewhat overstated by the authors. I also have some concerns when it comes to the methodology followed.

We thank the reviewer for their favorable overall assessment of the manuscript and their constructive suggestions. We have taken great care to address the concerns of the reviewer, summarized below and described in detail in response to the individual review comments, and believe that these modifications address the reviewer's concerns and strengthen the manuscript substantially:

- We provide extensive additional discussion of the existing literature and the specific novelty and import of our work.
- We conduct a new set of simulations allowing for future assessment of the additivity of single-species simulations and provide expanded discussion of the value of our multi-species simulation approach.
- We provide additional background on the precedent for using slab ocean simulations for studies of this kind, as well as for our statistical approach.
- We greatly expand the discussion of mechanisms driving the translation of atmospheric aerosol burdens from each regional emissions and of processes underlying the cloud response, as suggested by the reviewer, providing increased depth of analysis.
- We have corrected all other minor issues highlighted by the reviewer.

More specifically:

a) A key feature/finding, i.e. the stronger responses to forcing in extratropical versus tropical locations, is not discussed for the first time in the scientific literature. Shindell and Faluvegi (2009) had demonstrated and highlighted the fact that the location of aerosol and other forcings matters for global/regional temperature response in a coupled model (and did similarly for precipitation, in a subsequent paper in 2012), while there were earlier such indications as well in papers such as Forster et al. (2000). This past work which is highly relevant (especially the 2009 paper) is not mentioned in the introductory parts of the current manuscript, where existing work is discussed, but only briefly in the results section (L59). Breaking down the influence by region of emission has also been studied before (Aamaas et al. (2017), not cited in the manuscript; Collins et al. (2013), already cited); it may have been done using a two-step process (i.e. not with a single modeling framework), but these studies provided useful new insights, such as the fact that the influences may also vary with emission region within the same latitude band (e.g. European aerosols were found to be more important for global climate than East Asian aerosols in both studies, as is also found here). What is new in this and some other efforts that are currently ongoing in the community globally is the effort to separate effects from different regions in a seamless, atmosphere-ocean modeling approach (though see comment below regarding the use of a mixed-layer ocean). However, the above literature should have been discussed in the introduction, as currently the novelty of the study appears more pronounced than it actually is.

We appreciate the reviewer's recognition of the value of our novel methodology to "separate effects from different regions in a seamless, atmosphere-ocean modeling approach", and thank them for the opportunity to more thoroughly articulate the specific novelty and import of our work relative to the existing literature. We now expand discussion of existing work in the introduction of the revised manuscript, and remain confident that our work provides a unique perspective and valuable new insights to past work on this important topic.

We view the central novelty of our work to be the finding that identical aerosol emissions from major economies can result in substantially divergent global-scale temperature responses meriting consideration in scientific and policy analysis. Unlike past work, our simulation set-up achieves an ideal medium between clean quantification and decision-making relevance by allowing for a unique one-to-one comparison (cleanly interpretable in scientific and decision-making contexts) between aerosol emissions (the perturbation unit that is most directly influenced by societal decisions) in past, present, or projected future major emitting geopolitical regions (approaching the geographic delineations along which mitigation decisions and emissions accounting are conducted).

This assessment is uniquely enabled by our novel simulation set-up, perturbing identical aerosol emissions at the national scale, which has not—to our knowledge—been attempted before. This allows for several key findings not highlighted in other studies: that a unit of aerosol emissions in different countries can result in a 14x range in global-mean temperature effects, that countries have drastically different rates at which they feel the effects of their own aerosol emissions, and that the forcing efficacy of aerosols from individual major emitting economies differs by a factor of five. Our work also provides important new contributions to the scientific theory of aerosols' climate effects from its assessment of all steps of the process between emissions and temperature change, particularly its quantification of the role of radiative feedbacks, as enabled by our end-to-end modeling approach.

The community has made great progress in building a theoretical framework for understanding the importance of spatial heterogeneity in where aerosols are emitted. However, the few existing studies that attempt to map temperature responses to regionalized aerosol perturbations have substantial limitations for use in both decision-making and scientific contexts, limitations which we aim to address with our novel approach:

- Radiative forcings, atmospheric concentrations, or changes in the solar constant, which were used as the unit of perturbation in Forster et al. 2000, Shindell and Faluvegi, 2009, and Shindell et al., 2012 are not directly tied to societal decisions, as mitigation targets are generally set in terms of emissions, not in terms of radiative forcing.
- Latitude bands, which were the mode of regional delineation in Shindell and Faluvegi, 2009; Shindell, 2012, Shindell et al., 2012, Forster et al., 2000, and in certain steps of Aamaas et al., 2017, are not the geographic region along which mitigation decisions and emissions accounting are conducted.
- Historical emissions in different countries or scalings thereon, as assessed by Koch et al., 2007, Collins et al., 2013, Conley et al., 2018, Liu et al., 2018, and Aamaas et al., 2017, are unequal, and therefore cannot be used as straightforwardly to create a quantitative framework for assessing the relative importance of aerosol emissions from different regions in the context of climate metrics used in many scientific and policy frameworks.
- Many of these studies attempting to map the differential impacts of aerosols from individual regions do not do so through a single-model framework (e.g. Shindell, 2012, ACP, Collins et al., 2013, Aamaas et al., 2017), preventing the in-depth, mechanistic assessment of processes driving the divergence in response to aerosol that we provide here, as the reviewer notes. Additionally, while these methods provide a clever approach to estimate temperature responses with lower computational cost, they may compound biases by translating the initial aerosol signal through multiple uncertain scaling relationships to produce temperature response estimates, rather than directly modeling the temperature response as we do.
- Other studies assess processes translating regional emissions to radiative forcings or similar forcing metrics (e.g. Naik et al., 2007, Henze et al., 2012, Bond et al., 2011), but do not directly model or assess the temperature response, which we show here to be poorly predicted by the radiative forcing.

We now include a deeper description of the motivation for this work and its novelty in the introduction, as well as a more complete summary of existing work in this area to date and the ways in which this work presents novel advancements at L39-78 as follows:

“The scientific community has made great progress to date in building a theoretical framework for understanding the importance of the spatial heterogeneity in anthropogenic aerosol forcing. However, limitations remain in leveraging the existing literature to address key scientific and decision-making issues surrounding the differential role of aerosols emitted from different regions. Many studies

assessing the temperature response to regionally distinct aerosol perturbations use radiative forcing or atmospheric aerosol concentrations as their unit of regional perturbation^{18–20}. However, these units are not easily attributable to individual regions, as radiative forcing or atmospheric concentrations occurring in a given region may be attributable to aerosol emissions from well outside that region’s boundaries. This framing is additionally difficult to apply in decision-making contexts, as mitigation targets are generally set in terms of emissions rather than concentrations or radiative forcing. Several studies have looked at the effects of all aerosols within a given latitude band^{18–22}, but this is not a geographic delineation along which mitigation decisions, emissions accounting, or conjunct trends in aerosol emissions occur. Other studies have assessed the relative effects of historical emissions or scalings thereon in different countries^{16,22–25}. However, these emissions are unequal and therefore cannot be used straightforwardly to create a quantitative framework for assessing the relative importance of aerosol emissions from different regions in the context of the climate metrics used in many scientific and policy frameworks.

Here, we address a fundamental outstanding question underpinning both the consideration of aerosol emissions in climate policy and the analysis of their evolving role in global and regional climate change and climate sensitivity: can the temperature effects of a unit of aerosols emitted from any major emitting economy be considered equivalent? We analyze the relative climate effects of identical combined sulfate, black carbon, and organic carbon aerosol emissions – equivalent to China’s total annual year 2000 anthropogenic emissions¹ – sourced from 8 major past, present, or projected future emitting regions (Fig. 1a) in a global atmospheric general circulation model coupled to a mixed-layer ocean model (see Materials and Methods). These three aerosol species drive the vast majority of anthropogenic aerosol forcing over the historical period²⁶. They have historically varied in tandem at the national-scale, as economy-wide transitions have driven aerosol emissions changes over the historical period¹, and are projected to continue to do so in future^{27,28}. This novel design provides a crucial advancement by allowing for a unique one-to-one comparison (cleanly interpretable in scientific and decision-making contexts) between aerosol emissions (the perturbation unit that most directly corresponds to mitigation decision-making) in several past, present, or projected future major emitting geopolitical regions (approaching the geographic delineations along which mitigation decisions and emissions accounting are conducted). Our results demonstrate substantial inequalities in both the magnitude and spatial distribution of temperature effects of aerosol emissions located in different major emitting economies. These divergent temperature effects are fundamentally driven by differences in the degree to which each region’s emissions and the resulting distribution of radiative effects generate remote feedback processes, and reveal new challenges for understanding and addressing the global and regional climate influence of aerosols.

b) It is not convincing that applying perturbations to two aerosol species (sulfate and black carbon) that have such difference optical properties and TOA forcing is a good idea. These species may often be co-emitted, though there are many exceptions (to the extent that they are not exceptions). In fact, several of the most common BC measures that are being suggested for assisting climate change mitigation (for example those discussed in the Integrated Assessment for Black Carbon and Tropospheric Ozone by UNEP – see Box 5.1) do not involve large quantities of sulfur emissions. I would argue that having the efficacies (or temperature potentials) for different pollutants separately is a “cleaner” approach. If one preferred to design an alternative approach that has even more relevance to policy, the most suitable way to go would be to perturb emissions from different sectors in different regions, i.e. all the species emitted by separate sectors, and accounting for the “real” emission factors of different species in different sectors. Finally, I find the last sentence of the justification of this choice (i.e. L296-298) hard to understand.

We agree with the reviewer that there are a range of simulation set-ups that may be useful in assessing the importance of aerosols emitted from different regions. However, we believe that our current set-up provides important new perspectives that justify the perturbation of multiple species in tandem. We describe below the particular benefits of our multi-species approach, as well as our future plans to conduct complementary single-species simulations that will allow us to test the assumption of additivity that underlies single-species approaches to assessing aerosols’ climate impacts.

Although individual proposed mitigation strategies are intended to preferentially mitigate one species or another, the reality is that these species are emitted in tandem by the major economies at present. When viewed at a national-scale, these species have also been documented to vary in tandem with

time historically (e.g. Lamarque et al., 2013), as economy-wide transitions have driven aerosol emissions changes over the historical period, and are projected to continue to do so in future (Rao et al., 2017, Rogelj et al., 2014). Thus, while single species simulations are more straightforward to assess the output of and provide a cleaner theoretical framework, they do not generally map to the reality of large-scale emissions transitions.

A key novelty and motivation of this work is the ability to isolate the role of emissions location at the national-scale on the climate response to aerosols. A sectoral approach would not allow for the straightforward assessment of the importance of emissions location that our simulation set-up provides, as it is not clear that the emissions would be equivalent in each region in the set-up that the reviewer describes. Further, we view our simulation set-up as providing an optimal medium between idealization and applicability by imposing an easily interpretable perturbation that maps to observed present-day emissions in a major emitting economy without invoking uncertain assumptions about the future evolution of economic sectors in individual countries.

We agree that single-species simulations may be a useful variation in future. Indeed, we hope in future work to conduct a set of similar simulations with the individual aerosol species perturbed independently. However, the “cleaner” single species analysis suggested by the reviewer may not provide additive information on the aggregate effects of economy-wide aerosol transitions, particularly at the national scale. Although some studies have found reasonable additivity in the historical climate response to individual aerosol species in the global-mean, there are substantial non-linearities in regional responses (e.g. Jones et al., 2007, Ocko et al., 2014). These nonlinearities can be particularly pronounced in monsoonal regions, like India and China, in which the different vertical partitioning of aerosol radiative effects between the surface and atmosphere that occurs when absorbing and scattering aerosol species are emitted in tandem versus in isolation can strongly affect atmospheric stability and circulation (e.g. Persad et al., 2017). These nonlinearities could be particularly important for the degree to which regions feel the climate effects of their own aerosol emissions, a characteristic that we evaluate in this work. The all-species simulations presented in this manuscript will be an important component of testing the degree of additivity, and we hope to shed additional light on the role of these nonlinearities in the response observed in these simulations through future single-species simulations.

In preparation to address the separate additivity question through single species simulations, which we agree with the reviewer is a useful future extension, we have now redone all simulations to include identical organic carbon emissions. The resulting simulations, presented in the revised manuscript, now better capture the effects of economy-wide aerosol emissions. In the process of setting up the novel simulations, we identified minor logic errors in the original simulations affecting the precise equality of the regional emissions, and so do not include the original simulations alongside these revised simulations. However, the fact that the key findings—that the temperature response to identical emissions in different regions differs widely, that forcing efficacy differs by approximately a factor of five, that regions such as India feel the effects of their own emissions much more strongly, and that emissions from past emitters have stronger cooling potential than emissions from potential future emitters—are all either upheld or amplified in these new simulations gives us increased confidence in the credibility of these findings.

We have added a discussion of the above motivation for the multi-species approach at L347-361 (note that this update removes the previous L296-298 referenced by the reviewer):

“Although black carbon, sulfate, and organic carbon aerosol have somewhat opposing optical properties and global-mean top-of-atmosphere radiative effects, which can complicate analysis when they are co-emitted, we choose to include all of these species in each simulation. The species are co-emitted by many industrial processes, and their mitigation and growth often occur in tandem – a characteristic seen in their Representative Concentration Pathway and Shared Socioeconomic Pathway trajectories^{27,28}. Analyzing the climate effects of their simultaneous mitigation or growth is, therefore, likely to provide a better proxy for the climate effects of economy-wide transitions in aerosol emissions, as has been the historical norm and as is projected across the current suite of emissions scenarios used by the climate modeling and policy communities. Simulations that include only one aerosol species may not be additive, particularly in regional responses⁶³, reducing their utility in assessing the climate response to changes in multiple aerosol species simultaneously, as has been the dominant mode of aerosol emission change in reality. Future work will aim to illuminate the degree

of additivity in the multi-species response through comparison with a planned suite of complementary single species simulations.”

We also now highlight this framing of the interpretation of the simulation design in the context of economy-wide emissions transitions in the introduction (L65-68):

“These three aerosol species drive the vast majority of anthropogenic aerosol forcing over the historical period²⁶. They have historically varied in tandem at the national-scale, as economy-wide transitions have driven aerosol emissions changes over the historical period¹, and are projected to continue to do so in future^{27,28}”

Lamarque, J.-F., T. C. Bond, V. Eyring, C. Granier, A. Heil, Z. Klimont, D. Lee, et al. “Historical (1850–2000) Gridded Anthropogenic and Biomass Burning Emissions of Reactive Gases and Aerosols: Methodology and Application.” *Atmos. Chem. Phys.* 10, no. 15 (August 3, 2010): 7017–39. <https://doi.org/10.5194/acp-10-7017-2010>.

Rao, S. et al. Future air pollution in the Shared Socio-economic Pathways. *Global Environmental Change* 42, 346–358 (2017).

Rogelj, J. et al. Air-pollution emission ranges consistent with the representative concentration pathways. *Nature Clim. Change* 4, 446–450 (2014).

Jones, Andy, James M. Haywood, and Olivier Boucher. “Aerosol Forcing, Climate Response and Climate Sensitivity in the Hadley Centre Climate Model.” *Journal of Geophysical Research: Atmospheres* 112, no. D20.. <https://doi.org/10.1029/2007JD008688>.

Ocko, Ilissa B., V. Ramaswamy, and Yi Ming. “Contrasting Climate Responses to the Scattering and Absorbing Features of Anthropogenic Aerosol Forcings.” *Journal of Climate* 27, no. 14 (May 7, 2014): 5329–45. <https://doi.org/10.1175/JCLI-D-13-00401.1>.

Persad, Geeta G., David J. Paynter, Yi Ming, and V. Ramaswamy. “Competing Atmospheric and Surface-Driven Impacts of Absorbing Aerosols on the East Asian Summertime Climate.” *Journal of Climate*, August 17, 2017. <https://doi.org/10.1175/JCLI-D-16-0860.1>.

c) It is not clear to me whether the way that statistical significance of the results is assessed is appropriate in this case. It may be a method sometimes used in such analysis, but I do not believe it is entirely right. The authors show results as 60-year means, as I understand. What would then matter is how much the climate system varies on that timescale, i.e. 60 years. There are modes of variability that occur on multi-decadal timescales, and for those to be captured, one would need to have a picture of how the system (i.e. global climate in this model) could evolve on that timescale in an unforced situation. This is typically done by other performing multiple (or at least a few) ensemble members, or by extending (at least) the control simulation to a multiple of 60 years and then calculating the error from 60-year segments of that run. I am therefore wondering whether the method is adequate, and specifically whether uncertainty has been underestimated? For example, I was quite surprised to see such high confidence in cloud responses to aerosol forcings (even very remotely) in Fig. S6.

The statistical paradigm that the reviewer raises is more applicable to analyses in which the imposed perturbations are time evolving, rather than annually recurring, or in which the analysis is concerned with multi-decadal trends. The statistical approach and repeating year-to-year perturbation simulation design used here is widely used and accepted across modeling studies in the literature [e.g. Possner and Caldeira, 2018, Conley et al., 2018, Cao et al., 2017, Devaraju et al., 2015, Dietmüller et al, 2014, He et al., 2013, etc.]. Because the simulations contain repeating annual cycles of all forcings, the individual years of the simulation can be treated similarly to individual ensemble members (once autocorrelation between years is adjusted for, as we do following the methodology of Santer et al., 2000). The 60 year averages that we present, therefore, can be interpreted similarly to an ensemble average. The multi-decadal unforced variability is assessed in this approach, as the significance measure is calculated as a function of the inter-annual variability in the control simulation over the 60 year period. NCAR CESM is known to have a relatively low internal variability compared to other GCMs [Knutson et al, 2013], which may explain the reviewer’s surprise at the high confidence. Given the widespread use and acceptance of this statistical approach in the peer-reviewed literature for applications similar to our own and the absence of citations demonstrating its inadequacy, we have opted to maintain use of this statistical method in this version of the manuscript.

Cao, Long, Lei Duan, Govindasamy Bala, and Ken Caldeira. "Simultaneous Stabilization of Global Temperature and Precipitation through Cocktail Geoengineering." *Geophysical Research Letters* 44, no. 14 (July 24, 2017): 7429–37. <https://doi.org/10.1002/2017GL074281>.

Conley, A. J., D. M. Westervelt, J.-F. Lamarque, A. M. Fiore, D. Shindell, G. Correa, G. Faluvegi, and L. W. Horowitz. "Multimodel Surface Temperature Responses to Removal of U.S. Sulfur Dioxide Emissions." *Journal of Geophysical Research: Atmospheres* 123, no. 5 (n.d.): 2773–96. <https://doi.org/10.1002/2017JD027411>.

Devaraju, N., Govindasamy Bala, and Angshuman Modak. "Effects of Large-Scale Deforestation on Precipitation in the Monsoon Regions: Remote versus Local Effects." *Proceedings of the National Academy of Sciences* 112, no. 11 (March 17, 2015): 3257–62. <https://doi.org/10.1073/pnas.1423439112>.

Dietmüller, S., M. Ponater, and R. Sausen. "Interactive Ozone Induces a Negative Feedback in CO₂-Driven Climate Change Simulations." *Journal of Geophysical Research: Atmospheres* 119, no. 4 (n.d.): 1796–1805. <https://doi.org/10.1002/2013JD020575>.

He, Feng, Steve J. Vavrus, John E. Kutzbach, William F. Ruddiman, Jed O. Kaplan, and Kristen M. Krumhardt. "Simulating Global and Local Surface Temperature Changes Due to Holocene Anthropogenic Land Cover Change." *Geophysical Research Letters* 41, no. 2 (n.d.): 623–31. <https://doi.org/10.1002/2013GL058085>.

Knutson, Thomas R., Fanrong Zeng, and Andrew T. Wittenberg. "Multimodel Assessment of Regional Surface Temperature Trends: CMIP3 and CMIP5 Twentieth-Century Simulations." *Journal of Climate* 26, no. 22 (March 15, 2013): 8709–43. <https://doi.org/10.1175/JCLI-D-12-00567.1>.

Possner, Anna, and Ken Caldeira. "Geophysical Potential for Wind Energy over the Open Oceans." *Proceedings of the National Academy of Sciences* 114, no. 43 (October 24, 2017): 11338–43. <https://doi.org/10.1073/pnas.1705710114>.

d) Mixed-layer ocean model versions are certainly helpful as they involve shorter simulation times, and they often help isolate different oceanic influences (when compared to a full-depth ocean model), but in this case it would have been preferable to use a full-depth ocean model, as the full climate responses in a new equilibrium will have likely been modulated by ocean circulation changes - especially the way that heat is being redistributed across and between the hemispheres. The authors mention that "Responses in CAM5 coupled to a mixed-layer ocean have been shown to be well correlated with responses in CAM5 coupled to a fully dynamical ocean", and cite Bitz et al. (2012). However, Bitz et al. seem to have evaluated CAM4 and not CAM5 which was used here, so the above statement is a bit misleading. Furthermore, I could not locate where the cited paper suggests this correlation (although it is a long and detailed paper and I may have missed this).

We agree that fully coupled simulations are optimal for many applications. However, they are frequently computationally prohibitive for multi-regional assessments, such as what we are pursuing here. This is due both to the greater computational intensity of a fully interactive ocean model and to the additional length of simulations necessary to produce reliable signals with the higher variability induced by full coupling, as the reviewer acknowledges. Simulations using CAM5 coupled to a mixed-layer ocean have been widely used in the peer-reviewed literature [e.g. Gettelman, 2012; Modak et al., 2016], including to assess the climate response to aerosols [e.g. Ganguly et al., 2012, Clark et al., 2015].

The reviewer is correct that Bitz et al., 2012 assessed mixed-layer ocean versus full ocean coupling with CAM4 rather than CAM5, and we have modified the text to correct this error (L309-312). However, the simulations evaluated in Bitz et al., 2012 use the same mixed-layer ocean that we use here. It is expected that this similarity of mixed-layer versus full ocean coupled ocean simulations will be common to the CESM suite, and this has been used to motivate mixed-layer ocean simulations using CAM5 for applications similar to ours (Gettelman et al, 2012, see Section 2. Methodology).

Given these factors, we would argue that slab ocean simulations are an appropriate tool for this analysis. However, we do not exclude the future possibility of conducting fully coupled simulations of select regional aerosol emissions identified to be of particular interest through this work, which could allow assessment of the role of long timescale ocean circulation changes in modifying the results seen here.

We agree that it is important to fully articulate these considerations and have included the above discussion in the manuscript at L307-314 as follows:

“Simulations for this study were conducted in the National Center for Atmospheric Research Community Atmosphere Model 5 (NCAR CAM5), the atmospheric component of the Community Earth System Model 1⁵², coupled to a mixed-layer ocean. Mixed-layer coupling provides benefits in decreased computational intensity compared to full-ocean coupling, and has been found to lead to similar responses to full ocean coupling in the CESM model suite using an earlier version of the CESM atmospheric model⁵³. This similarity is expected to hold with use of CAM5⁵⁴. Simulations using CAM5 coupled to a slab ocean have been widely used in the peer-reviewed literature⁵⁴⁻⁵⁶, including to assess the climate response to aerosols^{57,58}.”

Gettelman, A., J. E. Kay, and K. M. Shell. “The Evolution of Climate Sensitivity and Climate Feedbacks in the Community Atmosphere Model.” *Journal of Climate* 25, no. 5 (September 13, 2011): 1453–69. <https://doi.org/10.1175/JCLI-D-11-00197.1>.

Clark, Spencer K., Daniel S. Ward, and Natalie M. Mahowald. “The Sensitivity of Global Climate to the Episodicity of Fire Aerosol Emissions.” *Journal of Geophysical Research: Atmospheres* 120, no. 22 (n.d.): 11,589-11,607. <https://doi.org/10.1002/2015JD024068>.

Ganguly, Dilip, Philip J. Rasch, Hailong Wang, and Jin-Ho Yoon. “Climate Response of the South Asian Monsoon System to Anthropogenic Aerosols.” *Journal of Geophysical Research: Atmospheres* 117, no. D13. Accessed June 25, 2018. <https://doi.org/10.1029/2012JD017508>.

Modak, Angshuman, Govindasamy Bala, Long Cao, and Ken Caldeira. “Why Must a Solar Forcing Be Larger than a CO₂ Forcing to Cause the Same Global Mean Surface Temperature Change?” *Environmental Research Letters* 11, no. 4 (2016): 044013. <https://doi.org/10.1088/1748-9326/11/4/044013>.

Other Comments:

L39-40: Any reason why other types of anthropogenic aerosols were excluded? Similarly (not referring to the same lines), why is only East Africa studied and not West Africa? And why was Eastern Europe not included, given that its emissions are a substantial fraction of European emissions?

We now include identical organic carbon aerosol emissions in the simulations to better capture economy-wide anthropogenic aerosol emissions (see response to Major Comment b). These three aerosols species account for the vast majority of present-day global anthropogenic aerosol emissions (e.g. Lamarque et al., 2010) and anthropogenic aerosol forcing over the historical period (Myhre et al., 2013).

The 8 geopolitical regions are chosen to sample locations that either have been, currently are, or are projected to become major emitting regions, while also providing a diversity of climatological conditions in which to test the response to regional aerosol emissions. Hence, we do not test regions immediately adjacent to each other with very similar climatological conditions, such as Eastern and Western Europe. Western Europe and the United States are past major emitting regions chosen to sample the response to aerosols emitted in Northern Hemisphere mid-latitude climate regions with different longitudinal locations and associated storm track regimes. India and China were chosen to represent current major emitting regions with two different monsoonal paradigms. Indonesia, Brazil, East Africa, and South Africa were chosen as projected potential future major emitting regions. Indonesia was chosen to test the impact of aerosol emitted within the deep convective western Pacific warm pool region. Brazil and East Africa were chosen to test the impact of aerosols emitted within different branches of the Intertropical Convergence Zone. South Africa was chosen to test the impact of Southern Hemisphere mid-latitude emissions.

Additional discussion of the motivation for the aerosol species used is now included at L65-68 and L347-361 (see also response to Major Comment b above).

The additional discussions of the motivation for the choice of regions has now been included in the Methods section at L334-344 as follows:

“These 8 geopolitical regions were selected to sample major past, present, or projected future emitting economies located in a range of climate regimes. Western Europe and the United States, major emitting regions over the historical period, were chosen to sample the response to aerosols emitted in Northern Hemisphere mid-latitude climate regions with different longitudinal locations and associated storm track regimes. India and China, current major emitting regions, were chosen to capture the response to two different monsoonal paradigms. The projected potential future emitting regions⁶², Indonesia, Brazil, East Africa, and South Africa, were chosen to assess the impact of aerosols emitted in the following respective climate regimes: the deep convective western Pacific warm pool region, the Pacific and Atlantic basin branches of the Intertropical Convergence Zone, and the Southern Hemisphere midlatitudes.”

Figure 1: Axes labels could be magnified.

We have increased the size of the axis labels on this and all other figures in the manuscript.

Discussion of Figure 2: Is there any dependence of the forcings on the background aerosols existing in different regions? That is worth some mention.

This section has been substantially updated (see below response to comment on L111-118) and now contains quantification of the relative importance of different mechanisms in driving the translation of the aerosol burdens into forcing. We find that most of the variance in ERF is explained by the rate of cancellation between the absorbing and scattering aerosol burdens produced by each regional emission (quantified by the global-mean ratio of total aerosol optical depth to absorbing aerosol optical depth, AOD/AAOD). The background aerosol fields could, in theory, impact the radiative environment that the additional aerosol burden generated by each regional emissions is interacting with. However, the regional change in aerosol burden in the perturbation is several times the control background aerosol in all of our simulations (240-480%, 590-750%, 310-830% increase for black carbon, sulfate, and organic carbon aerosol, respectively, depending on emissions region). Given how much larger the perturbation is than the control background aerosol in all cases, we do not expect that background aerosol will play a large role in differentiating the ERF produced by the regional emissions. The strong explanatory power of the AOD/AAOD variation also suggests that factors like differences in background aerosol are secondary to the differences in AOD/AAOD.

We have now added an acknowledgment of the theoretical role that background aerosol could play at L145-147, as part of our greatly expanded discussion of the relationship between burden and forcing, as follows:

“The remaining variance [in the relationship between burden and forcing] may be explained by 1) the radiative environment in which the aerosol population is interacting, created by regional differences in climatological cloud cover or background aerosol...”

Figure 2, right panel: “2” should be raised to a superscript.

This has been corrected.

L106: Why not also actually show the relationship between burden and forcing in a similar scatter plot?

We did not initially show the burden-forcing relationship because of the expected cancellation between the radiative effects of the black carbon burden and the sulfate burden. The total burden of aerosol would therefore not be expected to be a good predictor of the forcing.

We have now added substantial new analysis relating the aggregate optical depth of the aerosol burdens to the forcing. In order to accommodate the differing influence of the absorbing and scattering components of the aerosol burden on the radiative forcing we now regress the ratio of total aerosol optical depth to absorbing aerosol optical depth against the radiative forcing values for each region. This is now show in Figure 2a. We find that this explains roughly 60% of the variance in global-mean effective radiative forcing between regional emissions.

Additional discussion of this can be found in the response to the reviewer's comment on L111-118 below.

L108: This is referring to global imbalance, I presume? Please clarify.

Yes. We now take additional care to specify whether we are referring to global values or regional values throughout the manuscript.

L111: From Fig. 2, the difference seems to mainly be large for sulfate, not for BC.

This figure has now been replaced by the aggregate assessment of the ratio of total aerosol optical depth to absorbing aerosol optical depth versus effective radiative forcing. However, the spread in sulfate, black carbon, and organic carbon burdens resulting from each regional emission is comparable as a percent of the average burden change across the different regional emissions (57%, 56%, and 79%, respectively), and is now shown in Supplementary Figure 4.

L111-118: This discussion a bit rushed and hand-wavy. It is important to know which factor(s) may be more important here, at least for the example that the authors have picked (East African vs South African emissions).

We now greatly expand the assessment of the mechanisms driving the translation of the aerosol burdens into the different ERFs. We now quantify the ability of the cancellation between the absorbing and scattering aerosol burdens produced by each regional emission (quantified by the global-mean ratio of total aerosol optical depth to absorbing aerosol optical depth, AOD/AAOD) to explain the variation in effective radiative forcing (ERF) produced. We find that this explains approximately 60% of the variation in effective radiative forcing.

We also now include assessment of the role of other factors, like climatological cloud cover, on the effective radiative forcing produced, by assessing correlations with clear-sky ERF versus all-sky ERF. We find that the AOD/AAOD ratio has as much explanatory power for the clear-sky ERF ratio as for the all-sky ERF ratio indicating that the presence or absence of cloud does not substantially influence the translation of the atmospheric aerosol burdens into radiative forcing.

This expanded discussion can be found at L121-153 of the revised manuscript as follows:

“The total atmospheric aerosol burdens generated by the identical regional emissions are spread between 128–233 Gg of sulfate aerosol, 9.29–15.7 Gg of black carbon aerosol, and 23.1–54.2 Gg of organic carbon aerosol (Supp. Table 1). However, the atmospheric burdens of the individual aerosol species generated by each regional emission are largely uncorrelated with the regional emissions’ relative potency at changing global-mean temperature (Supplementary Figure 4). Thus, the disparity in temperature effects does not arise solely from a disparity in the atmospheric lifetime and resulting atmospheric burden of aerosols emitted from each region, but also through the differential generation of climate forcing and feedbacks by those burdens.

How do the aerosol burdens from each region’s emissions translate into radiative forcing, which in turn drives the global mean temperature change? The radiative effects of sulfate (a global-mean cooling agent), black carbon (a global-mean warming agent), and organic carbon (a global-mean cooling agent, though with minor shortwave absorbing properties) will counteract each other in driving the radiative forcing. This cancellation can be accommodated by using an aggregate measure of aerosol optical properties, such as aerosol optical depth. In order to capture the cancellation between the absorbing and scattering aerosol burdens, we calculate this as the ratio between the change in global-mean total aerosol optical depth and in global-mean absorbing aerosol optical depth caused by each regional emission.

The ratio of total to absorbing aerosol optical depth explains approximately 60% of the variance ($R=0.60$, $0.44-0.69$) in global-mean top-of-atmosphere effective radiative forcing (ERF) from each regional emission (Figure 2a). We calculate the ERF as the top-of-atmosphere radiative imbalance induced by the regional emission after the atmosphere and land surface has been allowed to respond (see Materials and Methods). The variance in ERF is thus largely explained by the rate of global-mean cancellation between the absorbing and scattering aerosol burdens resulting from each regional emission, arising from differences in aerosol mixing rates, relative altitudes, and other microphysical and radiative factors^{29,30}. The remaining variance may be explained by 1) the radiative environment in which the aerosol population is interacting, created by regional differences in climatological cloud cover or background aerosol; or 2) the particular cloud/convective environment in which the aerosols

are placed and the influence this has on the manifestation of the aerosols' indirect and semidirect effects on clouds^{31,32}. We find that the ratio of total aerosol optical depth to absorbing aerosol optical depth is approximately as well correlated with clear-sky ERF ($R=0.58, 0.31-0.61$) as it is with the total ERF, indicating that the presence or absence of climatological cloud does not substantially impact the translation of the atmospheric aerosol burdens into radiative forcing."

Figure 3: Suggest adding "aerosol" before "emissions".

We have made this modification.

L158-170: The surface albedo feedback influence seems to be given priority in the discussion (also in the abstract, in terms of order of mention), but the cloud feedback is more dominant (both its mean magnitude but also its absolute difference between e.g. the E. Africa and US or W. Europe simulations, based on Fig. 4).

We have now modified this discussion and the abstract to reflect the comparable explanatory power of the surface albedo feedback and cloud feedback in setting the differing efficacies.

See, e.g.:

L16-18: "We show that this behavior is fundamentally driven by differential excitement of radiative feedbacks from both cloud and sea ice responses remote from the aerosols themselves."

L194-196: The regional differences in the combined radiative gain from the cloud and surface albedo feedbacks (Fig. 4a) explains 84% of the variance in efficacy across regional emissions (Fig. 4b).

L190-197: Why would the Brazilian perturbation generate a similar ITCZ shift to the northern mid-latitude perturbations, when its forcing is mainly confined to the Southern Hemisphere? Similarly, why would E. African emissions lead to a similar response, e.g. over the Pacific, even though the emissions are primarily over the Equator. Sea-ice feedbacks on clouds themselves could be an explanation (i.e. hemispherically preferential sea ice feedbacks could affect the inter-hemispheric heating imbalance in a way that favours the specific ITCZ shift pattern), but there are sea-ice changes in the SH too.

Indeed, as we briefly referenced in the previous version of the manuscript at L196-197, we expect that the sea ice response may play a role in amplifying the ITCZ shift and associated cloud feedbacks. This amplification helps to explain the ITCZ shifts that appear to be non-correspondent with the particular location of the aerosol radiative effects. We now expand this discussion and include information on the total sea ice change in each hemisphere in response to each regional emission in Supplementary Figure 7. Even in the presence of Southern Hemispheric emissions, Arctic sea ice increases more strongly than Antarctic sea ice in all cases. This is likely attributable to the stronger overall climate sensitivity of the Arctic relative to the Antarctic (e.g. Goosse et al., 2018, Salzmann, 2017, Serreze et al., 2011). This common hemispheric imbalance in sea ice response drives comparable ITCZ shifts in response to many of the regional aerosol emissions.

We have included this expanded discussion at L222-233 as follows:

"In our simulations, the cloud feedbacks generated in several regions largely manifest through a north-south shift in tropical cloud cover (Supp. Fig. 6) associated with the intertropical convergence zone (ITCZ). This meridional ITCZ shift results from the large-scale atmospheric circulation adjusting to compensate for the hemispheric radiative imbalance induced by the localized aerosol forcing and its climate effects³⁹⁻⁴². This is likely amplified by the surface albedo feedback to each regional emission, which will generate its own hemispheric energy imbalance⁴³. Even in the presence of Southern Hemispheric emissions, Arctic sea ice increases more strongly than Antarctic sea ice in all cases (Supplementary Figure 7). This is likely attributable to the stronger overall regional climate sensitivity of the Arctic relative to the Antarctic⁴⁴⁻⁴⁶. This common hemispheric imbalance in sea ice response contributes to the comparable ITCZ shifts seen in response to many of the regional aerosol emissions."

Goosse, Hugues, Jennifer E. Kay, Kyle C. Armour, Alejandro Bodas-Salcedo, Helene Chepfer, David Docquier, Alexandra Jonko, et al. "Quantifying Climate Feedbacks in Polar Regions." *Nature Communications* 9, no. 1 (May 15, 2018): 1919. <https://doi.org/10.1038/s41467-018-04173-0>.

Salzmann, M. "The Polar Amplification Asymmetry: Role of Antarctic Surface Height." *Earth Syst. Dynam.* 8, no. 2 (May 18, 2017): 323–36. <https://doi.org/10.5194/esd-8-323-2017>.

Serreze, Mark C., and Roger G. Barry. "Processes and Impacts of Arctic Amplification: A Research Synthesis." *Global and Planetary Change* 77, no. 1 (May 1, 2011): 85–96. <https://doi.org/10.1016/j.gloplacha.2011.03.004>.

L210-213: Yes, useful point to make. But worth noting that climate models inherently include this redistribution of emissions in future scenarios driving them. Therefore this is not a missing process, but possibly it is not a fully appreciated one (e.g. IAMs do not commonly include such regional influences, as the authors mention earlier).

The reviewer's observation that full climate models will include this redistribution is correct, and we appreciate the opportunity to more clearly articulate this point. Many models used in assessment of policy implications, such as integrated assessment models or earth system models of intermediate complexity, underestimate this effect or exclude it entirely. For example, the Dynamic Integrated Climate-Economy (DICE) integrated assessment model, widely used in policy assessments, assumes time-constant translation of aerosol emissions into temperature effects (e.g. Nordhaus, 2014).

We now include this discussion at L249-257 as follows:

"While fully coupled climate models will capture the effects of these changes in overall rate of offsetting, the reduced form integrated assessment models (IAMs) and earth system models of intermediate complexity (EMICs) used in many impact assessment settings would fail to capture this signal. The widely used Dynamic Integrated Climate-Economy (DICE) integrated assessment model, for example, assumes a time-constant translation of aerosol emissions into temperature effects in calculating future global-mean climate change¹⁷. It would therefore overestimate the future offsetting effect of anthropogenic aerosol emissions, if their global distribution evolves toward lower potency regions as our results suggest."

Nordhaus, William. "Estimates of the Social Cost of Carbon: Concepts and Results from the DICE-2013R Model and Alternative Approaches." *Journal of the Association of Environmental and Resource Economists* 1, no. 1/2 (March 1, 2014): 273–312. <https://doi.org/10.1086/676035>.

L216: Suggest clarifying that this refers to global temperature change and radiative forcing.

We have added this clarification. We now take additional care to specify whether we are referring to global values or regional values throughout the manuscript.

L247-248: This seems somewhat contradictory to the picture described earlier in L13-24. Efficacies should surely matter even on a global scale, no? Or maybe I am misunderstanding something, in which case maybe this means that this could be clarified.

We intended here to contrast the large range in efficacies that we find for the individual regional aerosol emissions with the much smaller range in efficacies of radiative forcing resulting from global total concentrations of different forcers, as identified by Hansen et al. 2005 in their seminal work on forcing efficacy (c.f. Supplementary Table 1 with e.g. Hansen et al., 2005: Table 4). However, we have opted to remove this somewhat oblique reference from the revision of the manuscript. This sentence now reads (L291-293): *"Our results show that efficacy differences are pronounced for national-scale emissions changes—the scale at which climate accounting schemes and mitigation cost-benefit analyses will be undertaken."*

L258: See also the more recent study by Kasoar et al. (2016), which actually focused on regional emission impacts and explored causes of model diversity in the responses in detail.

We now also cite Kasoar et al., 2016 at what is now L301.

L299-305: How many years have been used for the ERF calculations? Please mention. The maps on Fig. S3 seem particularly noisy.

The fixed sea surface temperature simulations used to calculate the effective radiative forcing were run for 60 years. The final 40 years of the simulation was used for the ERF calculation. This contributes to the higher noise level, as identified by the reviewer, than the results derived from the mixed-layer ocean-coupled simulations, which include 60 years of data.

We have expanded the discussion of the fixed sea surface temperature simulations used to calculate ERF in the Methods section to include this information as follows.

L362-367: *“Effective radiative forcing (ERF) values are calculated using a suite of 9 CAM5 simulations formulated in the same way as the mixed-layer ocean simulations described above, but with the mixed-layer ocean and sea ice module replaced with a repeating annual cycle of observed year 2000 sea surface temperatures (SSTs) and sea ice coverage. These simulations are run for 60 years, the model equilibrates within the first 20 years, and the final 40 years of the simulation are used in the ERF calculation.”*

L391-394: *“Error ranges for the values in Figures 1, 2, and 4, Supplementary Figures 4 and 6, and Supplementary Table 1 are calculated as the standard error of the mean, using the final 60 years of each simulation’s data, with the exception of the ERF values in Supplementary Table 1, which use the final 40 years of the Fixed SST simulations’ data.”*

L398-399: *“Figure 5 is calculated using the final 40 years of the Fixed SST simulation data.”*

Aamaas, B., Berntsen, T. K., Fuglestedt, J. S., Shine, K. P., and Collins, W. J. (2017), Regional temperature change potentials for short-lived climate forcers based on radiative forcing from multiple models, *Atmos. Chem. Phys.*, 17, 10795-10809, <https://doi.org/10.5194/acp-17-10795-2017>.

Collins, W. J., Fry, M. M., Yu, H., Fuglestedt, J. S., Shindell, D. T., and West, J. J. (2013), Global and regional temperature-change potentials for near-term climate forcers, *Atmos. Chem. Phys.*, 13, 2471-2485, <https://doi.org/10.5194/acp-13-2471-2013>.

Forster, P. M. d. F., Blackburn, M., Glover, R. & Shine, K. P. (2000), An examination of climate sensitivity for idealised climate change experiments in an intermediate general circulation model, *Clim. Dyn.*, 16, 833–849.

Kasoar, M., Voulgarakis, A., Lamarque, J.-F., Shindell, D. T., Bellouin, N., Collins, W. J., Faluvegi, G., and Tsigaridis, K. (2016), Regional and global temperature response to anthropogenic SO₂ emissions from China in three climate models, *Atmos. Chem. Phys.*, 16, 9785-9804, <https://doi.org/10.5194/acp-16-9785-2016>.

Reviewer #3 (Remarks to the Author):

This paper investigates how global surface air temperature responds differently to aerosol emissions from different geographic regions. Of especial merit is the demonstration how (a) identical emission, when released in different regions, may lead to differences in (b) atmospheric burden, (c) radiative forcing, (d) feedback and (e) temperature response in order. The results have strong implications for assessment of historical forcing and policy making. I recommend publication after the following issues, mostly minor, are properly addressed.

We thank the reviewer for their favorable assessment of the manuscript and the impact of the findings, as well as their helpful suggestions for improvement. We have taken care to address all minor issues raised by the reviewer, which have provided valuable improvements to the revision.

One comment is that the uncertainties that arise in each of the above steps (a-b-c-d-e) need to be better noted in the paper. It would strengthen the paper to particularly evaluate the uncertainty in effective radiative forcing, which explains the majority of the global warming difference ($R^2=0.55$) but, from Fig S3, appears not explained by the burden difference. Hence, a key question is how the different ERFs result from the burdens. For instance, it is noticed that there is much remote effects and strong cloud adjustment in response to Indian emission, which leads to the small warming potential of the emission in this region - a key conclusion of the paper. The question is how much uncertainty is in this result? Note that it has been recognized that model-simulated adjusted forcing can be very uncertain judged from disparity among different models [e.g., Vial et al. 2013].

We appreciate the reviewers recognition of the value in identifying the role of steps a-e in generating the differences in temperature response to identical emissions. We have now added extensive additional assessment to more fully evaluate the uncertainties associated with each of these steps, which we summarize here and provide extended detail on below:

1. We now fully assess the mechanisms translating the aerosol burdens in effective radiative forcings and uncertainties therein, as requested by the reviewer, including the potential role of factors like climatological cloud cover in impacting the translation of burdens to forcings.
2. We now expand our analysis and discussion of interactions between the remote effects highlighted by the reviewer.
3. We now provide additional quantifications of uncertainty to numerous calculations throughout the manuscript.
4. We now replicate the central findings of the original manuscript in updated simulations that include identical organic carbon aerosol emissions (to address comments by Reviewer 2), demonstrating that the key conclusions are robust to formulation choices.

Additional details on each of these improvements are included below.

1. We now greatly expand the assessment of the mechanisms driving the translation of the aerosol burdens into the different ERFs, as requested by the reviewer.

We now quantify the ability of the cancellation between the absorbing and scattering aerosol burdens produced by each regional emission (quantified by the global-mean ratio of total aerosol optical depth to absorbing aerosol optical depth, $AOD/AAOD$) to explain the variation in effective radiative forcing (ERF) produced. We find that this explains approximately 60% of the variation in effective radiative forcing.

We include assessment of the role of other factors, like climatological cloud cover, on the effective radiative forcing produced, by assessing correlations with clear-sky ERF versus all-sky ERF. We find that the $AOD/AAOD$ ratio has as much explanatory power for the clear-sky ERF ratio as for the all-sky ERF ratio indicating that the presence or absence of cloud does not substantially influence the translation of the atmospheric aerosol burdens into radiative forcing.

This expanded discussion can be found at L121-153 of the revised manuscript as follows:

“The total atmospheric aerosol burdens generated by the identical regional emissions are spread between 128–233 Gg of sulfate aerosol, 9.29–15.7 Gg of black carbon aerosol, and 23.1–54.2 Gg of organic carbon aerosol (Supp. Table 1). However, the atmospheric burdens of the individual aerosol species generated by each regional emission are largely uncorrelated with the regional emissions”

relative potency at changing global-mean temperature (Supplementary Figure 4). Thus, the disparity in temperature effects does not arise solely from a disparity in the atmospheric lifetime and resulting atmospheric burden of aerosols emitted from each region, but also through the differential generation of climate forcing and feedbacks by those burdens.

How do the aerosol burdens from each region's emissions translate into radiative forcing, which in turn drives the global mean temperature change? The radiative effects of sulfate (a global-mean cooling agent), black carbon (a global-mean warming agent), and organic carbon (a global-mean cooling agent, though with minor shortwave absorbing properties) will counteract each other in driving the radiative forcing. This cancellation can be accommodated by using an aggregate measure of aerosol optical properties, such as aerosol optical depth. In order to capture the cancellation between the absorbing and scattering aerosol burdens, we calculate this as the ratio between the change in global-mean total aerosol optical depth and in global-mean absorbing aerosol optical depth caused by each regional emission.

The ratio of total to absorbing aerosol optical depth explains approximately 60% of the variance ($R=0.60$, $0.44-0.69$) in global-mean top-of-atmosphere effective radiative forcing (ERF) from each regional emission (Figure 2a). We calculate the ERF as the top-of-atmosphere radiative imbalance induced by the regional emission after the atmosphere and land surface has been allowed to respond (see Materials and Methods). The variance in ERF is thus largely explained by the rate of global-mean cancellation between the absorbing and scattering aerosol burdens resulting from each regional emission, arising from differences in aerosol mixing rates, relative altitudes, and other microphysical and radiative factors^{29,30}. The remaining variance may be explained by 1) the radiative environment in which the aerosol population is interacting, created by regional differences in climatological cloud cover or background aerosol; or 2) the particular cloud/convective environment in which the aerosols are placed and the influence this has on the manifestation of the aerosols' indirect and semidirect effects on clouds^{31,32}. We find that the ratio of total aerosol optical depth to absorbing aerosol optical depth is approximately as well correlated with clear-sky ERF ($R=0.58$, $0.31-0.61$) as it is with the total ERF, indicating that the presence or absence of climatological cloud does not substantially impact the translation of the atmospheric aerosol burdens into radiative forcing."

2. We also now provide further analysis of the factors controlling the cloud feedback to each regional aerosol emission and interactions between the different remote processes resulting from each regional emission. This includes an update of Supplementary Figure 7 to include additional information on the hemispheric imbalance in sea ice response and the role this plays in the cloud response. Even in the presence of Southern Hemispheric emissions, Arctic sea ice increases more strongly than Antarctic sea ice in all cases. This is likely attributable to the stronger overall climate sensitivity of the Arctic relative to the Antarctic. This common hemispheric imbalance in sea ice response drives comparable ITCZ shifts in response to many of the regional aerosol emissions.

We have included this expanded discussion at L222-233 as follows:

"In our simulations, the cloud feedbacks generated in several regions largely manifest through a north-south shift in tropical cloud cover (Supp. Fig. 6) associated with the intertropical convergence zone (ITCZ). This meridional ITCZ shift results from the large-scale atmospheric circulation adjusting to compensate for the hemispheric radiative imbalance induced by the localized aerosol forcing and its climate effects³⁹⁻⁴². This is likely amplified by the surface albedo feedback to each regional emission, which will generate its own hemispheric energy imbalance⁴³. Even in the presence of Southern Hemispheric emissions, Arctic sea ice increases more strongly than Antarctic sea ice in all cases (Supplementary Figure 7). This is likely attributable to the stronger overall regional climate sensitivity of the Arctic relative to the Antarctic⁴⁴⁻⁴⁶. This common hemispheric imbalance in sea ice response contributes to the comparable ITCZ shifts seen in response to many of the regional aerosol emissions."

3. We now provide more complete calculations of uncertainty on all analysis in the manuscript, including:

- The addition of uncertainty bars to the radiative feedback calculations show in Supplementary Figure 6, including the responses in clouds and sea ice, as highlighted by the reviewer
- The addition of uncertainty ranges on all R values cited in the manuscript, using jackknife resampling to assess the sensitivity of our R value calculations to individual regions.

- Expanded discussion of statistical calculations used to assess all uncertainties in the Materials and Methods section at L391-402.

4. We have now redone all simulations to include identical organic carbon aerosol emissions, in addition to black carbon and sulfate aerosol, in response to comments by Reviewer 2. The resulting simulations, presented in the revised manuscript, now better capture the effects of economy-wide aerosol emissions. In the process of setting up the new simulations, we identified minor logic errors in the original simulations affecting the precise equality of the regional emissions, and so do not include the original simulations alongside these revised simulations. However, the fact that the key findings—that the temperature response to identical emissions in different regions differs widely, that forcing efficacy differs by approximately a factor of five, that regions such as India feel the effects of their own emissions much more strongly, and that emissions from past emitters have stronger cooling potential than emissions from potential future emitters—are all either upheld or amplified in these new simulations gives us increased confidence in the credibility of these findings.

Line 13-14. Note the greenhouse gas forcing is also heterogeneous [Huang et al. 2016]. This ought to be noted when comparing aerosol and greenhouse gas forcings and their climate responses.

We thank the reviewer for pointing out this subtlety. We now acknowledge this at L14-17 as follows: *“Although carbon dioxide radiative forcing has some spatial structure associated with the climatological radiative environment [Huang et al., 2016], the historical spatial distribution of aerosol forcing has been shown to generate a larger transient and equilibrium global-mean climate response than equivalent amounts of long-lived greenhouse gas forcing...”*

Line 17. What does “projection” mean here?

By “greater projection onto Northern Hemisphere land and polar regions”, we intended to refer to the fact that historical aerosol radiative forcing is more spatially correlated with Northern Hemisphere land and polar regions, which has been hypothesized to be a primary driver of its greater efficacy at generating global-mean climate response compared to more homogeneous forcings (e.g. Marvel et al., 2016). We have modified this sentence as follows for clarity (L16-19):

“the historical spatial distribution of aerosol forcing has been shown to generate a larger transient and equilibrium global-mean climate response than equivalent amounts of long-lived greenhouse gas forcing, as a result of historical aerosol forcing’s greater spatial coincidence with Northern Hemisphere land and polar regions⁹⁻¹¹.”

Line 39. What does “functionally” mean?

We have removed “functionally” from this sentence. It has now been updated and clarified as follows (L57-60): *“Here, we address a fundamental outstanding question underpinning both the consideration of aerosol emissions in climate policy and the analysis of their evolving role in global and regional climate change and climate sensitivity: can the temperature effects of a unit of aerosols emitted from any major emitting economy be considered equivalent?”*

Line 320. The feedback results can be validated, e.g., by checking the radiation closure [e.g., see example given by Huang et al. 2017] and/or checking the sensitivity to the kernel dataset used (note multiple sets are available).

We thank the reviewer for this suggestion and now include additional discussion of the validity of the kernels used.

The radiative kernels used in this study are generated in the same climate model (i.e. NCAR CESM-CAM5) as the simulations themselves [see Pendergrass et al., 2018], and are therefore the most appropriate for calculating radiative kernels in our model simulations [e.g. Soden et al., 2008; Shell et al., 2008], though we recognize that radiative kernels generated in other climate models are available.

Pendergrass et al., 2018 validate the NCAR CESM-CAM5 radiative kernels used here via radiative closure techniques as in Huang et al., 2017 and find: “Errors in global-mean radiative flux change range from 0.1 to 0.8 Wm⁻², while global-mean absolute errors range from 0.7 to 1.4 Wm⁻².”

We have included the above information at L374-377, including an updated reference to Pendergrass et al., 2018, where the assessment of radiative closure may be found:

“Use of radiative kernels generated within the same climate model as the simulations to which they are applied, as is the case here, is generally considered to be best practice for reducing errors associated with differences in radiative transfer codes between climate models^{64,68}”.

Pendergrass, Angeline G., Andrew Conley, and Francis M. Vitt. “Surface and Top-of-Atmosphere Radiative Feedback Kernels for CESM-CAM5.” *Earth System Science Data* 10, no. 1 (February 21, 2018): 317–24. <https://doi.org/10.5194/essd-10-317-2018>.

Figures. Some fonts in the labels and legends are too small.

We have now increased the label and legend sizes in all figures in the manuscript.

References

Huang, Y., X. Tan and Y. Xia, (2016), Inhomogeneous radiative forcing of homogeneous greenhouse gases, *J. Geophysical Res.-Atmosphere*, 121, doi: 10.1002/2015JD024569.

Huang, Y., Y. Xia and X. Tan, (2017), On the pattern of CO₂ radiative forcing and poleward energy transport, *J. Geophys. Res.-Atmosphere*, doi: 10.1002/2017JD027221.

Vial, J., J.-L. Dufresne, and S. Bony, 2013: On the interpretation of inter-model spread in CMIP5 climate sensitivity estimates. *Climate Dyn.*, 41, 3339–3362, doi:10.1007/s00382-013-1725-9.

Reviewer #2 (Remarks to the Author):

The authors have done a very good job addressing and/or thoroughly and extensively discussing my comments. I am satisfied with the additions/modifications made and responses given, and I am happy to recommend publication of the manuscript.

Reviewer #3 (Remarks to the Author):

The authors have satisfactorily addressed all my comments. I recommend publication.